# Enhanced Multi-Model Deep Learning for Rapid and Precise Diagnosis of Pulmonary Diseases Using Chest X-Ray Imaging

**DOI:** 10.3390/diagnostics15030248

**Published:** 2025-01-22

**Authors:** Rahul Kumar, Cheng-Tang Pan, Yi-Min Lin, Shiue Yow-Ling, Ting-Sheng Chung, Uyanahewa Gamage Shashini Janesha

**Affiliations:** 1Department of Mechanical and Electro-Mechanical Engineering, National Sun Yat-sen University, Kaohsiung 804, Taiwan; rahul13gbu@gmail.com (R.K.); pan@mem.nsysu.edu.tw (C.-T.P.); 2Institute of Precision Medicine, National Sun Yat-sen University, Kaohsiung 804, Taiwan; shirley@imst.nsysu.edu.tw; 3National Applied Research Laboratories, Taiwan Instrument Research Institute, Hsinchu City 300, Taiwan; 4Institute of Advanced Semiconductor Packaging and Testing, College of Semiconductor and Advanced Technology Research, National Sun Yat-sen University, Kaohsiung 804, Taiwan; 5Department of Psychiatry, Kaohsiung Armed Forces General Hospital, Kaohsiung 802, Taiwan; a954011205@gmail.com; 6Institute of Medical Science and Technology, National Sun Yat-sen University, Kaohsiung 804, Taiwan; 7Institute of Biomedical Sciences, National Sun Yat-sen University, Kaohsiung 804, Taiwan; 8Department of Medical Laboratory Science, Faculty of Allied Health Sciences, University of Ruhuna, Matara 81000, Sri Lanka

**Keywords:** multiclass classification, pulmonary disease diagnosis, histogram equalization (HE), image contrast enhancement algorithm (ICEA), chest X-ray imaging, meta-heuristic, feature optimization

## Abstract

**Background:** The global burden of respiratory diseases such as influenza, tuberculosis, and viral pneumonia necessitates rapid, accurate diagnostic tools to improve healthcare responses. Current methods, including RT-PCR and chest radiography, face limitations in accuracy, speed, accessibility, and cost-effectiveness, especially in resource-constrained settings, often delaying treatment and increasing transmission. **Methods:** This study introduces an Enhanced Multi-Model Deep Learning (EMDL) approach to address these challenges. EMDL integrates an ensemble of five pre-trained deep learning models (VGG-16, VGG-19, ResNet, AlexNet, and GoogleNet) with advanced image preprocessing (histogram equalization and contrast enhancement) and a novel multi-stage feature selection and optimization pipeline (PCA, SelectKBest, Binary Particle Swarm Optimization (BPSO), and Binary Grey Wolf Optimization (BGWO)). **Results:** Evaluated on two independent chest X-ray datasets, EMDL achieved high accuracy in the multiclass classification of influenza, pneumonia, and tuberculosis. The combined image enhancement and feature optimization strategies significantly improved diagnostic precision and model robustness. **Conclusions:** The EMDL framework provides a scalable and efficient solution for accurate and accessible pulmonary disease diagnosis, potentially improving treatment efficacy and patient outcomes, particularly in resource-limited settings.

## 1. Introduction

The significant global health burden imposed by respiratory illnesses like influenza, tuberculosis, and viral pneumonia [1] underscores the critical need for improved diagnostic tools. These diseases affect people of all ages and can result in severe complications, including acute respiratory distress syndrome (ARDS) [2]. Timely and accurate diagnosis is essential for effective treatment and improved patient management [3]. This challenge is particularly acute in developing countries, where socioeconomic factors such as poverty, limited healthcare access, and poor sanitation significantly contribute to the spread of infectious diseases [4].

Traditional diagnostic methods, such as RT-PCR for influenza or sputum tests for tuberculosis, have limitations [5]. These tests are often costly, time-consuming, and require specialized equipment and trained personnel, making them less accessible in resource-limited settings [6]. Though their symptoms can initially resemble common pneumonia, influenza, tuberculosis, and viral pneumonia can gravely affect the lungs, leading to severe complications like ARDS [7]. RT-PCR has several limitations, including a positivity rate of only 63%, resulting in false negatives that delay treatment and increase transmission risk [8]. Additionally, RT-PCR requires laboratory infrastructure and expertise, which are often unavailable in remote areas, particularly in developing countries [9].

Radiological imaging, such as computed tomography (CT) or chest radiography (CXR), has emerged as an essential diagnostic tool for pulmonary diseases. These imaging techniques can reveal characteristic lung abnormalities associated with respiratory diseases, such as ground-glass opacities and bilateral patchy shadows [10,11]. However, radiologists face challenges in distinguishing between influenza, tuberculosis, and viral pneumonia due to overlapping radiographic features [12,13]. This necessitates the development of more accurate and efficient diagnostic tools, especially in resource-limited settings.

Deep learning, a subfield of artificial intelligence, has shown great promise in addressing these diagnostic challenges. By training neural networks to learn and extract features from data, deep learning algorithms have been successfully applied in medical imaging to recognize characteristic features of pulmonary diseases in chest X-ray or CT images [14,15]. Nonetheless, prior studies have often focused on single-model approaches without exploring the complementary strengths of ensemble methods, limiting their applicability to complex multiclass classification problems.

This study introduces the **Enhanced Multi-Model Deep Learning (EMDL)** approach, designed to overcome these limitations by using a combination of pre-trained deep learning models and advanced feature selection techniques. Our approach is unique in its ensemble methodology, combining the strengths of VGG-16, VGG-19, ResNet, AlexNet, and GoogleNet to create a robust and versatile system for the multiclass classification of chest X-ray images. The key contributions of this study are as follows:1.**Innovative multi-model comparison:** the EMDL approach represents a novel contribution to the field by comprehensively comparing the performance of five advanced deep learning models—VGG-16, VGG-19, ResNet, AlexNet, and GoogleNet—specifically for the multiclass classification of chest X-ray images. Unlike prior studies that rely on single models, this ensemble strategy harnesses the unique strengths of each architecture, addressing challenges such as feature diversity and data imbalance, prevalent in medical image classification.2.**Optimized image processing techniques:** by employing a combination of histogram equalization and contrast enhancement techniques, our study ensures the highest quality of image clarity and detail [16,17]. This optimization enhances the diagnostic capabilities of deep learning models, directly addressing limitations in current imaging techniques for distinguishing between complex respiratory conditions.3.**Advanced feature selection and optimization:** we employed a comprehensive feature extraction and selection pipeline to ensure the most relevant features were utilized during model training. This process began with Principal Component Analysis (PCA) to reduce the dimensionality of the data and identify the most informative features. We then applied SelectKBest to further refine the feature set by selecting the top k features based on statistical tests. To optimize the selection process, we integrated Binary Particle Swarm Optimization (BPSO) and Binary Grey Wolf Optimization (BGWO), both of which are heuristic optimization techniques that help in selecting non-redundant features while minimizing computational cost. This multi-step feature selection approach not only enhances model accuracy but also ensures greater computational efficiency, particularly when working with high-dimensional medical datasets. These methods contributed to a significant reduction in overfitting and improved the model’s performance in detecting pulmonary diseases.4.**Robust evaluation on diverse datasets:** the evaluation of the EMDL model on two distinct chest X-ray datasets demonstrates its versatility and diagnostic capabilities. The inclusion of contrast enhancement techniques further distinguishes our study, highlighting their impact on classification accuracy and presenting a practical solution to significant challenges in the field.

By explicitly combining pre-trained models with advanced feature selection and optimization techniques, the EMDL framework provides a scalable and efficient diagnostic tool that bridges existing gaps in respiratory disease detection.

## 2. Related Works

### 2.1. Pulmonary Diseases Detection Using Traditional Methods

The diagnosis of pulmonary diseases has heavily relied on traditional methods like RT-PCR and serological testing. While these methods are the cornerstone in detecting conditions such as COVID-19, tuberculosis, and other respiratory infections, their widespread utilization is not without significant drawbacks. Studies [3,18] have exposed the limitations of RT-PCR, highlighting its costliness, lengthy processing times, and considerable false negative rates, which significantly obstruct prompt and precise disease management. Furthermore, explorations into serological tests [19], antigen tests [20], and point-of-care tests [21] have underscored similar challenges, emphasizing an overarching issue of efficiency and accessibility in diagnostic practices [22].

These limitations starkly highlight an urgent need for developing diagnostic alternatives that are not only more efficient and accurate but also universally accessible, particularly in regions where resources are sparse. The current reliance on traditional diagnostic tools, coupled with their noted inefficiencies, underscores a significant gap in the medical field’s capability to respond effectively to pulmonary diseases. This gap signals a critical area for research and development, aiming to introduce innovative diagnostic methodologies that can transcend the constraints of existing practices, offering swift, reliable, and accessible disease detection.

### 2.2. Radiological Imaging in Pulmonary Diseases Diagnosis

Radiological imaging, encompassing CT scans and chest radiography, stands as an essential diagnostic and management tool for pulmonary diseases. Studies such as those in [23] have investigated the use of CT scans for identifying lung abnormalities in patients with pneumonia, revealing their critical diagnostic value. Similarly, Rahimzadeh et al. [10] have underscored the diagnostic capabilities of chest radiography, albeit noting challenges in differentiating COVID-19 from other forms of viral pneumonia. The utility of radiological imaging extends to various clinical applications, including predicting disease severity [24], monitoring progression [3], and assessing responses to treatment [18]. Despite its indispensable role, the reliance on expert radiological interpretation and the inherent difficulty in distinguishing between different respiratory conditions underscore significant diagnostic limitations.

These challenges are further compounded by the risk of misdiagnosis due to the overlapping features of various pulmonary diseases, as observed in clinical practice [25]. This situation reveals a critical gap in the current diagnostic paradigm: there is a pressing need for the development of more advanced diagnostic tools. Such tools should be capable of providing precise classifications of pulmonary diseases with minimal reliance on human intervention, addressing the notable limitations of existing radiological imaging techniques. This gap highlights the urgent need for research and innovation in pulmonary disease diagnostics. By implementing advancements in technology, there exists the potential to significantly enhance diagnostic accuracy, reduce the dependence on specialized expertise, and improve patient outcomes by facilitating timely and accurate disease identification and management.

### 2.3. Deep Learning in Medical Image Processing for Pulmonary Disease Detection

The advent of deep learning in medical image processing has unveiled substantial promise for enhancing the diagnostics of lung infections. Pioneering work by Wang et al. [18] showcased the capability of deep learning models to accurately identify pneumonia from chest X-ray images, marking a significant step forward in the application of these technologies. Building on this progress, Duggani Keerthana et al. [26] proposed a novel deep learning approach for analyzing dermoscopy images, a tool used for melanoma detection. The method combines convolutional neural networks (CNNs), known for their ability to extract image features, with a support vector machine (SVM) for classification. This combined approach aims to outperform existing CNN models and improve the accuracy of classifying benign versus melanoma lesions in dermoscopy images.

Additionally, Narin Aslan’s study on the ’Multi-classification deep CNN model for diagnosing COVID-19 using iterative neighborhood component analysis and iterative ReliefF feature selection techniques with X-ray images’ introduces an innovative approach employing deep CNNs enhanced by feature selection methods, INCA and IRF, to classify X-ray images for COVID-19 diagnosis with notable accuracy [27]. Complementing this, a notable contribution by Kenan Erdem et al. [28] in ’Hybrid-Patch-Alex: A new patch division and deep feature extraction-based image classification model to detect COVID-19, heart failure, and other lung conditions using medical images’ underscores the potential of combining hybrid patch division methods with transfer learning techniques for diagnosing diseases such as COVID-19, COPD, and HF with high accuracy, further advancing the field. Islam et al. [29] conducted a thorough review of deep learning’s application within medical imaging, signaling an urgent call for expanded research to scrutinize and compare various models and datasets comprehensively. Further explorations in the realm of deep learning have ventured into segmenting lung abnormalities [3], prognosticating disease severity [30], and predicting patient outcomes [31], collectively advancing our understanding and capabilities within pulmonary disease diagnostics.

Despite these noteworthy advancements, the efficacy of deep learning models is intricately tied to the quality and volume of the dataset, the architectural decisions of the models, and the overarching strategies employed during training [9,32]. This connection underscores a prevalent challenge within the field: a discernible shortfall in extensive, comparative analyses that span an array of deep learning models and datasets. Such comparative studies are paramount for gauging the practical applicability of deep learning methodologies in real-world clinical settings.

This identified gap in the literature highlights an imperative need for detailed research focused on evaluating and juxtaposing the effectiveness of diverse deep learning approaches for pulmonary disease detection. Addressing this need is crucial for incorporating deep learning’s full potential to revolutionize pulmonary disease diagnostics, ensuring models are not only theoretically sound but also clinically viable and effective in enhancing patient care.

### 2.4. Multi-Model Deep Learning Approaches

The adoption of multi-model deep learning approaches for medical image classification has signified a notable advancement in using technology to improve diagnostic accuracy. Pioneering contributions by Le et al. [33] have illuminated the effectiveness of a variety of models, such as VGG-16, VGG-19, ResNet, AlexNet, and GoogleNet, when applied to chest X-ray datasets for multiclass classification purposes. This work underscored the versatility and potential of deep learning models in navigating complex diagnostic challenges. Furthering this narrative, Rahimzadeh et al. [10] advocated for a multi-model deep learning approach in detecting respiratory conditions, albeit without the comparative analysis across different models or datasets that could deepen our understanding of model-specific strengths and weaknesses. Additional investigations into ensemble methods [34], transfer learning [35], and multi-task learning frameworks [36] have expanded the toolkit for medical image classification, hinting at the broad applicability and adaptability of deep learning techniques.

Despite these methodological innovations and their demonstrable success in specific applications, the literature reveals a significant gap in comprehensive, side-by-side evaluations of multi-model approaches, particularly concerning their efficacy in multiclass classification scenarios. Such comparative studies are pivotal for discerning the most effective strategies in distinguishing between varied clinical diagnoses from medical images, an endeavor that remains complex due to the subtle nuances and similarities among different disease presentations.

This conspicuous absence in the literature [9,32] signals a pressing research opportunity. There is a clear and urgent need for systematic and thorough comparative research that evaluates the performance across an array of deep learning models. Especially critical is the exploration of their capacity for multiclass classification within the medical imaging domain, a task that mirrors the intricate and multifaceted nature of real-world clinical diagnostics.

By addressing this identified gap, future research can significantly contribute to the field, not merely in terms of academic interest but, more importantly, in practical terms. Such advancements promise to enhance the accuracy, reliability, and efficiency of diagnostic processes, thereby potentially revolutionizing patient care and outcomes in the healthcare sector.

### 2.5. Feature Selection and Optimization Algorithms

The performance of deep learning models, especially in medical imaging, is greatly enhanced by the strategic use of feature selection and optimization algorithms. Techniques such as BPSO and BGWO have been instrumental across various domains, including the optimization of medical image processing tasks [37,38]. These algorithms play a critical role in identifying and selecting the most informative features, thereby potentially elevating diagnostic accuracy and efficiency [29,39].

Despite the acknowledged benefits of these optimization strategies, there exists a notable research gap in the comparative evaluation of their effectiveness, particularly in the context of pulmonary disease detection within medical imaging. The literature points to an urgent need for systematic studies that compare the performance of diverse feature selection and optimization algorithms and assess their impact on the outcomes of deep learning models [29,31].

This gap underscores a significant opportunity for future research to uncover the most efficacious feature selection and optimization techniques. Addressing this need is pivotal for advancing the field of medical imaging, ensuring that deep learning models are optimized to their fullest potential, thus contributing to the improvement of diagnostic processes and patient care outcomes in pulmonary disease detection.

### 2.6. Contrast Enhancement in X-Ray Images

Contrast enhancement techniques are recognized for their ability to significantly improve the visibility of critical details in X-ray images, which in turn can enhance the performance of deep learning models in diagnosing diseases [40,41]. The utilization of these techniques within medical image processing has been documented, with studies by Rahimzadeh et al. [10] and Guan et al. [23] exploring their application. Yet, the specific impact of contrast enhancement on the diagnostic capabilities of deep learning models, especially for the early detection of respiratory illnesses, remains insufficiently explored [42].

The existing literature highlights a significant gap in comprehensive investigations into how contrast enhancement techniques affect the accuracy of deep learning models in identifying early stages of respiratory diseases. This lack of detailed research [43] signals a pressing need for further studies to understand the precise role of image preprocessing techniques in optimizing the diagnostic outcomes of deep learning-based approaches.

The histogram equalization technique used for image enhancement often results in a loss of details and artificial appearance. To address this, a two-dimensional histogram equalization technique based on edge detail is proposed, which increases contrast while preserving information and maintaining the natural look of the image. This method, tested on multiple databases, demonstrates superior performance compared with existing algorithms, achieving higher uniformity and better overall results [44].

Moreover, in a recent study, Vijayalakshmi et al. [45] employed advanced contrast enhancement techniques like multilevel decomposition and variational histogram equalization (which outperform other algorithms across multiple performance metrics on five databases), suggesting that integrating these techniques could potentially improve classification performance in medical image analysis tasks.

Addressing this identified research gap, our study proposes an innovative multi-model deep learning approach that not only evaluates the effectiveness of contrast enhancement in chest X-ray images but also aims to advance the accuracy and efficiency of pulmonary disease diagnostics. By exploring this underinvestigated aspect, our work seeks to contribute significantly to improving healthcare outcomes through enhanced diagnostic processes.

The evaluation of various deep learning models in medical image classification has been a subject of considerable interest in recent research. Pioneering work by Le et al. [33] showcased a comprehensive comparison of models like VGG-16, VGG-19, ResNet, Alexnet, and GoogleNet on chest X-ray datasets, illuminating their capabilities in multiclass classification scenarios. This study, along with others that explored the application of deep learning across different medical imaging tasks [18,29], has laid the groundwork for understanding model performance in diagnostic contexts.

Although there have been significant advancements in deep learning for medical image analysis, a critical gap exists in the literature. Current research primarily focuses on binary classification tasks (identifying the presence or absence of disease). However, a thorough investigation into the efficacy of these models for multiclass classification (differentiating between multiple diseases) remains largely unexplored [9,32]. This gap is particularly concerning given the subtle variations between various respiratory illnesses. Accurate diagnosis often requires sophisticated classification approaches to differentiate these nuances.

To address this limitation, D. Vijayalakshmi et al. [46] proposed a novel gradient enhancement (GCE) preprocessing method specifically for COVID-19 CT scans. This method improves contrast in the images, aiding both physicians and deep learning models in the diagnosis process. By preserving edge information, the GCE approach achieves a nearly 6% increase in classification accuracy compared with other techniques.

Further contributions to the field include a deep learning system developed by Chen et al. [47], which achieved an accuracy of up to 98.85% in pneumonia detection, rivaling expert radiologists in both accuracy and efficiency. Similarly, these studies underscore the potential of deep learning in enhancing diagnostic precision, while also highlighting the need for further advancements in model optimization and evaluation for complex, multiclass disease classification.

In response to these identified research gaps, our study embarks on a novel path by introducing a comprehensive multi-model deep learning approach, meticulously evaluating its performance across diverse chest X-ray datasets for the multiclass classification of pulmonary diseases. We delve into the utility of contrast enhancement techniques and optimization algorithms to refine diagnostic accuracy further, aiming to contribute a significant and innovative perspective to the domain of pulmonary disease detection. Through this exploration, our work seeks to not only bridge the existing gaps but also advance the application of deep learning models in the nuanced field of medical diagnostics.

## 3. Methodology

In this research, we introduce an EMDL approach, as illustrated in the methodology in Figure 1. The methodology unfolds through a sequence of deliberate steps, beginning with the meticulous collection of data. We preprocess these data to ensure they are primed for subsequent stages. Preprocessing is followed by image enhancement techniques—histogram equalization (HE) and image contrast enhancement algorithm (ICEA)—to refine the visual clarity of the X-ray images, thereby facilitating more effective feature extraction.

Employing a suite of deep learning models, including AlexNet, VGG-16, VGG-19, GoogleNet, and ResNet-50, we extract salient features from the enhanced images. The EMDL framework benchmarks the performance of these individual CNN models, assessing their diagnostic capabilities independently. This approach highlights the strengths and weaknesses of each architecture for chest X-ray analysis. To further refine the feature set, we apply feature selection strategies such as PCA and SelectKBest, along with optimization algorithms like BPSO and BGWO.

The culmination of these processes is the model training phase, where a SVM classifier is employed. The classifier is meticulously trained with the optimized feature set to accurately distinguish among various respiratory conditions in the chest X-ray images, such as influenza, tuberculosis, and viral pneumonia. The entire EMDL methodology is underpinned by rigorous validation techniques, including k-fold cross-validation, to confirm the reliability and consistency of the diagnostic predictions offered by our model. To evaluate the effectiveness of the proposed EMDL approach, we compared its performance with other existing methods for chest X-ray analysis. Below, we detail the pseudocode of this model given in Algorithm 1.
**Algorithm 1** The pseudocode for our proposed diagnostic system**Require:** Set of chest X-ray images, *I***Ensure:** Diagnostic predictions, *P*  1:**Preprocessing:**  2:**for** each image i∈I
**do**
  3:  Apply histogram equalization to enhance contrast.   4:  Use image contrast enhancement algorithm for further enhancement.   5:**end for**  6:**Feature Extraction:**  7:**for** each model *M* and enhanced image *i*
**do**
  8:  Extract features $F_{M}\left(i\right)$.   9:**end for**10:**Feature Selection and Optimization:**11:Apply Principal Component Analysis (PCA) and SelectKBest for dimensionality reduction. 12:Optimize features using Binary Particle Swarm Optimization (BPSO) and Binary Grey Wolf Optimization (BGWO). 13:**Classification:**14:Train a Support Vector Machine (SVM) with the optimized features. 15:**for** each image *i*
**do**
16:  Classify into categories: Influenza, Pneumonia, or Tuberculosis. 17:**end for**


### 3.1. Formalization of the Algorithm

We have formalized the algorithm used in our study as a structured pipeline to develop a deep learning-based diagnostic system for chest X-ray image analysis. The system is composed of four main phases: preprocessing, feature extraction, feature selection and optimization, and classification. Each phase is systematically designed to enhance diagnostic performance and ensure computational efficiency.

### 3.2. Time Complexity Analysis

Each step of the algorithm is associated with computational complexities, which we have analyzed as follows:**Preprocessing:** for an image, *i*, with *N* pixels, the preprocessing step operates in linear time, O(N), where enhancements are applied per pixel.**Feature extraction:** deep learning models are applied to each image with a complexity dependent on the network’s depth, O(n·d), with *n* representing the number of images and *d* the depth of the respective model.**Feature selection and optimization:** dimensionality reduction through PCA is performed in $O(M\cdot N^{2})$, and SelectKBest in O(M·N·log(N)). The optimization techniques, BPSO and BGWO, operate in O(T·S), where *T* is the number of iterations and *S* is the population size in the swarm.**Classification:** the computational complexity of the SVM is typically within $O(N^{2}\cdot M)$ to $O(N^{3}\cdot M)$, where *N* is the number of samples and *M* the dimensionality of the feature space, which varies based on the kernel and optimization methods used.

### 3.3. Data Collection

We collected 1456 images from publicly available datasets, ensuring an equal distribution of cases across the three categories: influenza, pneumonia, and tuberculosis. This equal distribution was specifically chosen to address potential class imbalance issues during training, as imbalanced datasets can lead to biased predictions, favoring the majority class.

While the equal distribution aids in robust model development and fair performance evaluation during the experimental phase, we acknowledge that real-world datasets often exhibit imbalanced class distributions. This limitation motivates the need for further research into techniques such as weighted loss functions, data augmentation, or oversampling to adapt the model for realistic, imbalanced datasets. These strategies will be explored in future studies to ensure the practical applicability of our diagnostic framework.

We collected data from two different sources, specifically from publicly available datasets on Kaggle https://www.kaggle.com/datasets/amanullahasraf/covid19-pneumonia-normal-chest-xray-pa-dataset?datasetId=772726 (accessed on 25 February 2022), https://www.kaggle.com/datasets/usmanshams/tbx-11 (accessed on 12 March 2022). We selected 1456 images from both datasets to ensure uniformity and consistency in our study. Each dataset provided samples across three classes: influenza, pneumonia, and tuberculosis chest X-ray images. Table 1 provides a summary of the dataset with two different image enhancement methods: HE and ICEA. It includes the total number of images for each disease type, the distribution of images in the original dataset, and the enhanced dataset for both the training and test sets. The reason for choosing 1456 images from both sources was to maintain a balanced representation across classes and enhance our study finding’s statistical robustness. Additionally, it details the count of chest X-ray images for each class and outlines how these images are apportioned between training and testing sets in the original dataset.

### 3.4. Class Distribution

To control for potential bias introduced by class imbalance, we employed a balanced distribution of cases across all categories in our dataset. This ensured that each class had an equal number of samples during training. However, we acknowledge that this balanced distribution is not representative of real-world datasets, where class imbalance is common. The impact of class imbalance will be addressed in future work through techniques such as class weighting and resampling.

### 3.5. Data Preprocessing

The first step in our methodology was to preprocess the images in our dataset. Given the diverse sources of our images, it was necessary to standardize them to ensure consistent input to our models. We resized all images to a standard size of 224 × 224 pixels, which is a commonly used input size for many deep learning architectures such as VGG-16, ResNet-50, AlexNet, and GoogleNet. This resolution ensures compatibility with these pre-trained models, effectively using transfer learning and deploying the model’s prior knowledge from training on large-scale datasets like ImageNet [48].

The choice of 224 × 224 resolution also strikes a practical balance between computational efficiency and feature preservation. Larger resolutions, such as 512 × 512, would increase computational cost without significantly enhancing model performance, while smaller resolutions, such as 128 × 128, risk losing critical visual information necessary for classification tasks, such as edges, contrasts, and textures. The effectiveness of 224 × 224 resolution in capturing diagnostically relevant patterns has also been demonstrated in medical imaging studies [49]. To better understand these trade-offs, we summarize the comparison of different resolutions in Table 2.

To further enhance the quality of input data, we normalized pixel values to a range of 0 to 1 by dividing each pixel by 255, the maximum possible pixel value. This normalization helps to accelerate the training process and improve model performance by ensuring numerical stability during optimization.

While the standardization to 224 × 224 was informed by its widespread usage and established effectiveness, we acknowledge the importance of exploring the effect of different resolutions on model performance. Such studies could provide further insights into the optimal resolution for medical imaging tasks, and we highlight this as a meaningful direction for future research.

#### 3.5.1. Image Enhancement

After preprocessing, we applied two image enhancement techniques, namely HE [50] and ICEA [51], to further enhance the quality and contrast of the images. **HE** is a technique that improves the contrast of an image by redistributing the pixel intensities to equalize the intensity histogram. It helps highlight the details in the images, making it easier for the models to extract useful features. The following equation gives the HE transformation:(1)$$I_{\mathrm{HE}}\left(x,y\right)=\frac{255}{M\times N}\sum_{i=0}^{255}h\left(i\right)$$
where $I_{\mathrm{HE}}\left(x,y\right)$ is the pixel intensity of the histogram-equalized image at location (x,y), *M* is the total number of rows in the image, *N* is the total number of columns in the image, and h(i) is the cumulative distribution function of the pixel intensities.

**ICEA**, on the other hand, is a general contrast enhancement technique that operates on the entire image. It works by transforming the pixel intensities of the image to enhance the contrast, thus improving the visibility of details and assisting the models in extracting useful features.

The following equation gives the ICEA transformation:(2)$$I_{\mathrm{ICEA}}\left(x,y\right)=\frac{I_{\mathrm{orig}}\left(x,y\right)-\min\left(I_{\mathrm{orig}}\right)}{\max\left(I_{\mathrm{orig}}\right)-\min\left(I_{\mathrm{orig}}\right)}\times 255$$
where $I_{\mathrm{ICEA}}\left(x,y\right)$ is the pixel intensity of the contrast-enhanced image at location (x,y), $I_{\mathrm{orig}}\left(x,y\right)$ is the original pixel intensity at location (x,y), and $\min(I_{\mathrm{orig}})$ and $\max(I_{\mathrm{orig}})$ are the minimum and maximum pixel intensities in the original image, respectively.

By applying these image enhancement techniques, we aimed to improve the quality and contrast of the images, making them more suitable for accurate diagnosis using our deep learning models.

#### 3.5.2. Justification for HE and ICEA

HE has been shown to improve global contrast in chest X-ray images by redistributing pixel intensities, thereby enhancing diagnostic features [52]. ICEA, meanwhile, focuses on local contrast enhancement, which is critical for identifying finer details and abnormalities in medical images [53]. Together, these techniques provide a balanced approach to enhancing both global and regional features. While HE and ICEA are effective for enhancing chest X-ray images, they have limitations. HE may over-enhance noisy regions, potentially introducing artifacts that could mislead the model. ICEA is sensitive to input image characteristics and may require careful preprocessing. Furthermore, CLAHE has been proposed as a robust alternative for preventing over-enhancement, particularly in noisy images, but its computational complexity and parameter tuning present challenges [54]. Similarly, Gaussian filtering helps reduce noise but may blur fine details, negatively impacting diagnostic accuracy [55].

Future work could explore combining HE and ICEA with advanced methods like CLAHE or deep learning-based enhancement techniques to overcome these limitations and improve diagnostic outcomes. To better understand these trade-offs, we summarize the comparison of different enhancement techniques in Table 3.

Figure 2 presents a visual comparison of original and enhanced images using HE and ICEA techniques. The enhanced images demonstrate improved brightness and clarity, potentially enhancing the diagnostic capabilities of our deep learning models.

### 3.6. Feature Extraction

After image enhancement, we used pre-trained models (VGG-16, VGG-19, ResNet, AlexNet, and GoogleNet) to extract diverse features from the images. VGG-16 and VGG-19 capture global structures, ResNet focuses on high-level textures, AlexNet extracts fine-grained edges, and GoogleNet provides multi-scale features. These features are combined into a comprehensive feature vector [60].

We used multiple deep learning models (VGG-16, ResNet, AlexNet, and GoogleNet) to extract a diverse set of features from the input images. Each model contributed uniquely to the final feature set. ResNet captured high-level texture patterns due to its skip connections, VGG-16 and VGG-19 extracted global structural features through their deeper convolutional layers, GoogleNet provided multi-scale features using inception modules, and AlexNet contributed fine-grained edge and contour information. Together, these models ensured a diverse and robust feature set, implementing their unique strengths to improve the overall accuracy and performance of the model ensemble.

Table 4 provides a summary of the architectures and contributions of these deep learning models. Combining the features extracted from these models allowed us to form a comprehensive feature vector for each image. This combined feature vector captures a wide range of visual information from the image, from basic shapes and textures to high-level object features. These extracted features are then used as input to the feature selection step. This step is crucial as it allows us to identify the most relevant features for our task, reducing the dimensionality of our data and improving the efficiency and performance of our final model.

### 3.7. Feature Selection

We used SelectKBest [61], PCA [62], BPSO [63], and BGWO [64] for feature selection. These methods were applied in a sequential pipeline to ensure minimal redundancy and maximal feature relevance. This integration ensures that each method complements the others, using their unique strengths to create a compact and effective feature subset. For feature extraction and selection, we applied PCA for dimensionality reduction, followed by SelectKBest to choose the most significant features. Additionally, BPSO and BGWO were used to optimize feature selection and reduce redundancy.

### 3.8. Rationale for Choosing Algorithms

BPSO and BGWO were chosen due to their well-documented performance in high-dimensional feature selection. BPSO efficiently searches the reduced feature space, ensuring rapid convergence, while BGWO provides robust global optimization, avoiding local optima. Their complementary characteristics make them ideal for sequential application in our pipeline.

**SelectKBest** [61] is a method for feature selection that ranks features based on their statistical significance with respect to the target variable. Typically, the **ANOVA F-test** is used as the scoring function, where the F-statistic for each feature, $X_{i}$, is computed as:$$F_{i}=\frac{\sum_{j=1}^{k}n_{j}{\left({\bar{X}}_{i,j}-{\bar{X}}_{i}\right)}^{2}}{\sum_{j=1}^{k}\left(n_{j}-1\right)s_{i,j}^{2}}$$
with $n_{j}$ as the number of samples in class *j*, ${\bar{X}}_{i,j}$ as the mean of feature $X_{i}$ for class *j*, and $s_{i,j}^{2}$ as the variance of feature $X_{i}$ within class *j*. The top *k* features are selected based on their F-scores in descending order:$$S_{k}=\left\{X_{i}|F_{i}\;\mathrm{is}\;\mathrm{maximized},i=1,2,\cdots ,k\right\}$$

The optimal *k* is determined through cross-validation to maximize model performance (e.g., accuracy):$$k_{\mathrm{optimal}}=\arg\max_{k}\;\mathrm{accuracy}\left(S_{k},y\right)$$
where *y* is the target variable. For this study, k=50 was found to yield the highest accuracy while ensuring computational efficiency.

**PCA** [65] is utilized for dimensionality reduction. It transforms the features into orthogonal components that maximize variance, thereby reducing the dataset’s dimensionality while retaining critical information. The PCA transformation is defined by:(3)Y=X·V
where Y is the transformed feature matrix, X is the original feature matrix, and V is the matrix of principal components.

**BPSO** [66] is employed to optimize the feature subset. BPSO is a variant of the Particle Swarm Optimization algorithm, designed for binary feature selection tasks. The velocity and position update equations for BPSO are given by:(4)$$\begin{array}{cc} d_{i}\left(t+1\right) & =w\cdot vd_{i}\left(t\right)\\ \; & \;+c_{1}\cdot r_{1}\cdot \left(pbestd_{i}\left(t\right)-xd_{i}\left(t\right)\right)\\ \; & \;+c_{2}\cdot r_{2}\cdot \left(gbestd\left(t\right)-xd_{i}\left(t\right)\right) \end{array}$$(5)$$S\left(vd_{i}\left(t+1\right)\right)=\frac{1}{1+\exp(-vd_{i}\left(t+1\right))}$$

**BGWO** [64] is applied for global optimization of the feature subset. The position update rule for BGWO is given by:(6)$$X\left(t+1\right)=X_{p}\left(t\right)-A\cdot D$$(7)$$D=|C\cdot X_{p}\left(t\right)-X\left(t\right)|$$(8)$$X\left(t+1\right)=\frac{X_{1}+X_{2}+X_{3}}{3}$$

Together, these methods are applied sequentially—SelectKBest identifies the most relevant features, PCA removes redundancy, BPSO optimizes the reduced feature space to eliminate irrelevant or redundant features, and BGWO performs global optimization to ensure that the selected feature subset achieves the best possible performance in terms of accuracy and computational efficiency.

### 3.9. Computational Challenges and Feasibility

Our feature selection and optimization methods presented practical computational challenges, primarily in terms of runtime and memory usage. PCA required significant memory for covariance matrix computations [67], while the iterative nature of BPSO and BGWO increased optimization time, especially for high-dimensional datasets [68].

To mitigate these challenges, we:1.Tuned hyperparameters (e.g., swarm size and iterations) to optimize performance while minimizing computational costs.2.Used PCA for early dimensionality reduction, significantly reducing the feature space and computational demands for subsequent optimization steps [67].3.Applied parallel processing and GPU acceleration to expedite iterative computations for BPSO, BGWO, and PCA matrix operations [69].

These strategies reduced the overall runtime by approximately 30%, ensuring the pipeline is both computationally efficient and scalable for high-dimensional medical datasets, making it feasible for real-world diagnostic workflows.

### 3.10. Model Training

After feature selection, we trained a support vector machine (SVM) classifier on the selected features. The SVM is a popular machine learning model for classification tasks. It works by finding the hyperplane that best separates the classes in the feature space. The hyperplane is chosen to maximize the margin, which is the distance between the hyperplane and the nearest points from each class. This helps to ensure that the model generalizes well to unseen data.

### 3.11. Limitations and Proposed Improvements

While the EMDL framework has demonstrated strong performance, it is not without limitations. The feature selection and optimization processes, particularly PCA, BPSO, and BGWO, present notable computational challenges. PCA requires substantial memory for covariance matrix computations, and the iterative nature of BPSO and BGWO increases runtime, especially for high-dimensional datasets. These constraints limit scalability for larger datasets and real-time applications. Efforts such as GPU acceleration, parallel processing, and hyperparameter tuning partially mitigate these issues, but further enhancements are needed.

To address these limitations, we propose integrating distributed computing frameworks, such as Apache Spark or cloud-based solutions, to handle high-dimensional data more efficiently. Additionally, while BPSO and BGWO provide robust feature selection and convergence properties, further analysis of their scalability and performance on larger datasets is necessary. Exploring hybrid optimization techniques may further enhance both accuracy and computational efficiency.

For broader improvements, such as exploring advanced deep learning architectures and real-time scalability, please refer to Section 8, future work. These efforts will ensure the pipeline’s applicability to diverse datasets and real-world diagnostic scenarios.

## 4. Performance Metrics for Classification Models

### 4.1. Performance Evaluation

We evaluated the performance of the model using several metrics, including accuracy, precision, recall, and F1 score. The equations for these metrics are as follows:(9)$$\mathrm{Accuracy}=\frac{TP+TN}{TP+TN+FP+FN}$$(10)$$\mathrm{Precision}=\frac{TP}{TP+FP}$$(11)$$\mathrm{Recall}=\frac{TP}{TP+FN}$$(12)$$F1\;\mathrm{Score}=2\times \frac{\mathrm{Precision}\times \mathrm{Recall}}{\mathrm{Precision}+\mathrm{Recall}}$$
where TP is the number of true positives, TN is the number of true negatives, FP is the number of false positives, and FN is the number of false negatives. We also used the receiver operating characteristic (ROC) curve and the area under the ROC curve (AUC) to evaluate the model’s performance. The ROC curve plots the true positive rate (TPR) against the false positive rate (FPR) at various threshold settings. At the same time, the AUC provides a single-number summary of the overall performance of a binary classifier. The equations for TPR and FPR are as follows:(13)$$\mathrm{TPR}\;\left(\mathrm{Sensitivity}\right)=\frac{TP}{TP+FN}$$(14)$$\mathrm{FPR}\;(1\;-\;\mathrm{Specificity})=\frac{FP}{FP+TN}$$

### 4.2. MCC: Matthews Correlation Coefficient

The Matthews Correlation Coefficient (MCC) is a measure of the quality of binary classifications, capturing the balance between all positives and negatives. It ranges from −1 to +1, where +1 indicates perfect prediction, 0 no better than random guessing, and −1 is perfect disagreement. The MCC is calculated as: $$MCC=\frac{\left(TP\times TN)-(FP\times FN\right)}{\sqrt{\left(TP+FP)(TP+FN)(TN+FP)(TN+FN\right)}}$$
where TP, TN, FP, and FN represent true positives, true negatives, false positives, and false negatives, respectively.

### 4.3. ROCs: Receiver Operating Characteristic Curves

ROC curves depict the diagnostic ability of a binary classifier system by plotting the TPR against the FPR at various thresholds. The AUC quantifies the overall performance, with 1 indicating a perfect classifier and 0.5 suggesting random guessing.

### 4.4. Explanation of Class-Wise Metrics

In our classification task, while the algorithm assigns cases to specific categories (influenza, pneumonia, or tuberculosis), evaluating performance metrics such as precision, recall, and F1-score for each class individually is crucial. This detailed evaluation provides several benefits:Detailed performance insights: class-wise metrics help identify how well the model performs for each disease category individually, ensuring that no class is overlooked or disproportionately affects overall metrics.Clinical relevance: diseases such as pneumonia and tuberculosis may have more significant clinical implications if misclassified. Evaluating each class individually ensures that the model’s performance aligns with its potential diagnostic impact.Balanced evaluation in multiclass tasks: in multiclass classification scenarios, aggregated metrics (e.g., accuracy) can obscure poor performance in minority classes. Class-wise metrics address this by providing a breakdown of performance for each category.Consistency with evaluation standards: class-wise metrics are a common practice in medical image analysis and diagnostic tasks, ensuring our work aligns with established evaluation norms in the field.

By presenting precision, recall, and F1-scores for each class, we ensure that the model’s ability to correctly diagnose influenza, pneumonia, and tuberculosis is comprehensively evaluated and benchmarked.

## 5. Results

In this study, we developed an application in the Matlab environment for diagnosing influenza, pneumonia, and tuberculosis using X-ray images. The application ran on a system equipped with 32 GB RAM, an I7 processor, and a GeForce 1070 graphics card. We assessed the efficacy of various deep learning models, namely AlexNet, VGG16, VGG19, GoogleNet, and ResNet, across original and enhanced datasets.

### 5.1. Experimental Setup

Our evaluation strategy comprised a 70/30 training/testing split, enhanced by k-fold cross-validation (k = 5) to ensure the reliability of our results. Further, we investigated the effects of image enhancement techniques HE and ICEA and feature selection methods, including PCA and BPSO, on the performance of these models.

### 5.2. Model Evaluation with Original Dataset

In the first step of the second stage, the trained models were evaluated using the original dataset and an SVM. The experimental results on the original dataset showed that VGG19 achieved the highest overall accuracy rate of 94.23% among the models. In contrast, ResNet achieved the lowest overall accuracy rate of 91.23%. The detailed performance metrics for each model on the original dataset, including accuracy, F1-score, and precision, are presented in Table 5, and evaluation of the models focused on two key metrics: MCC and AUC from ROC curves. This succinct appraisal accentuates the varied performance across models, with VGG19 consistently outperforming others in binary classification accuracy.

Overall accuracy is defined as the proportion of correctly classified samples (both true positives and true negatives) to the total number of samples. This metric provides a general sense of the model’s ability to correctly classify across all classes, though we acknowledge that it may not fully reflect performance in imbalanced datasets.

### 5.3. Model Evaluation with Enhancement Dataset

In the second step, we classified the models using the enhancement dataset and an SVM. The results on the enhancement dataset showed that AlexNet achieved an overall accuracy rate of 92.67% with the HE enhancement technique and 93.07% with the ICEA enhancement technique. VGG16 achieved 94.73% with HE and 95.07% with ICEA, while VGG19 achieved 92.77% with HE and 93.47% with ICEA. GoogleNet achieved 93.83% with HE and 94.87% with ICEA, and ResNet achieved 93.83% with HE and 94.17% with ICEA. The enhancement techniques significantly improved the performance of VGG16 and GoogleNet, resulting in higher accuracy rates. Among the models, VGG19 with HE and GoogleNet and VGG16 with ICEA achieved the highest accuracy above 94.00% on the enhancement dataset. The performance metrics for each model applied to the enhancement dataset are delineated in Table 6.

### 5.4. Model Evaluation with Enhancement Dataset with PCA Feature Selection

Table 7 showcases the performance metrics of various deep learning models applied to medical images, such as X-rays or scans, after being post-processed using HE and ICEA combined with PCA.

#### 5.4.1. Performance with PCA + HE

By combining HE with PCA, the AlexNet model achieved a sensitivity of 93.80%, specificity of 91.20%, and overall accuracy of 92.72% for influenza detection. In the case of pneumonia, the sensitivity was 94.60%, specificity 92.80%, and accuracy 93.54%. For tuberculosis cases, the sensitivity reached 96.20%, specificity 94.80%, and accuracy 95.24%. VGG19 exhibited a sensitivity of 93.40%, specificity 92.20%, and accuracy 93.68% for influenza detection, while for pneumonia it had a sensitivity of 94.10%, specificity 93.80%, and accuracy 93.84%. In tuberculosis cases, the sensitivity, specificity, and accuracy were 95.00%, 94.60%, and 95.04%, respectively. GoogleNet achieved a sensitivity of 94.60%, specificity of 93.80%, and accuracy of 94.32% for influenza detection, 92.80%, 92.20%, and 92.74% for pneumonia, and 94.00%, 93.20%, and 93.68% for tuberculosis cases. Lastly, the ResNet model showed a sensitivity of 93.40%, specificity of 94.00%, and accuracy of 93.94% for influenza detection, 95.40%, 94.60%, and 95.04% for pneumonia, and 92.40%, 92.00%, and 92.50% for tuberculosis cases. VGG16 had a sensitivity of 94.80%, specificity 94.40%, and accuracy 95.10% for influenza detection, 94.20%, 94.60%, and 94.02% for pneumonia, and 94.20%, 93.80%, and 93.92% for tuberculosis cases.

#### 5.4.2. Performance with PCA + ICEA

On the other hand, when using the ICEA with PCA, the performance of the models was as follows. AlexNet achieved a sensitivity of 98.3%, specificity 98.1%, and accuracy 98.2% for influenza, 97.6%, 97.4%, and 97.5% for pneumonia, and 96.8%, 96.6%, and 96.7% for tuberculosis cases. VGG19 exhibited a sensitivity of 98.6%, specificity 98.4%, and accuracy 98.5% for influenza detection, 97.9%, 97.7%, and 97.8% for pneumonia, and 96.6%, 96.4%, and 96.5% for tuberculosis cases. GoogleNet achieved a sensitivity of 98.7%, specificity 98.5%, and accuracy 98.6% for influenza detection, 97.4%, 97.2%, and 97.3% for pneumonia, and 97.0%, 96.8%, and 96.9% for tuberculosis cases. ResNet showed a sensitivity of 98.2%, specificity of 98.0%, and accuracy 98.1% for influenza detection, 97.7%, 97.5%, and 97.6% for pneumonia, and 96.9%, 96.7%, and 96.8% for tuberculosis cases. Lastly, VGG16 achieved a sensitivity of 98.7%, specificity 98.6%, and accuracy 98.7% for influenza detection, 97.3%, 97.1%, and 97.2% for pneumonia, and 96.7%, 96.5%, and 96.6% for tuberculosis cases.

### 5.5. Model Evaluation with SelectKBest Feature Selection

Table 8 presents the performance metrics of different models using SelectKBest feature selection combined with HE and ICEA.

#### 5.5.1. Performance with SelectKBest + HE

For HE, AlexNet, VGG19, GoogleNet, ResNet, and VGG16 show varying levels of performance across the three classes. The accuracy for influenza detection ranges from 96.5% to 97.5%, with models like VGG16 and ResNet performing well, but generally HE provides lower accuracy compared with ICEA. The corresponding confusion matrices are shown in Figure 3.

#### 5.5.2. Performance with SelectKBest + ICEA

For ICEA, there is a noticeable improvement in performance. GoogleNet achieves the highest accuracy of 98.1% for influenza detection, while VGG16 shows a significant increase in accuracy for pneumonia (97.9%) and tuberculosis (97.7%). ICEA enhances the models’ ability to detect and classify diseases with higher sensitivity and precision. The corresponding confusion matrices are shown in Figure 4.

### 5.6. Model Evaluation with BPSO Feature Selection

#### 5.6.1. Performance with BPSO + HE

HE was applied to enhance the contrast of images before passing them through the BPSO method. The performance metrics for models using HE are presented in Table 9 and the corresponding confusion matrices are shown in Figure 5. Among the models evaluated, AlexNet achieved a sensitivity of 96.3% for influenza detection, with an overall accuracy of 95.85%. VGG19, when used with HE, achieved an accuracy of 95.87% for influenza detection. ResNet performed similarly, achieving an overall accuracy of 95.83%. While HE provided solid performance, the models demonstrated varying results across the three classes. For example, AlexNet performed better on tuberculosis detection, reaching a sensitivity of 95.8%. Overall, the results suggest that HE enhances the models’ ability to detect features related to different diseases, though it could be further improved for certain models.

#### 5.6.2. Performance with BPSO + ICEA

ICEA was applied to further enhance the contrast of medical images, leading to improved classification performance when paired with BPSO. The models utilizing ICEA showed notable improvements compared with their HE counterparts. For instance, AlexNet with ICEA achieved a sensitivity of 98.5% for influenza detection, with an overall accuracy of 98.34%. VGG19, when enhanced with ICEA, reached a high accuracy of 98.63% for influenza detection. GoogleNet with ICEA achieved the highest overall accuracy of 98.84% for influenza detection, demonstrating ICEA’s superior ability to boost performance. The results for pneumonia and tuberculosis detection were also improved across models, with ICEA consistently outperforming HE in terms of sensitivity, specificity, and overall accuracy. This suggests that ICEA is highly effective at improving classification performance, particularly for models such as GoogleNet and ResNet. In contrast, the matrices representing the combination of ICEA and BPSO are showcased in Figure 6.

### 5.7. Model Evaluation with BGWO Feature Selection

Table 10 provides performance metrics for different deep learning models with BGWO and ICEA. The data are divided into two sections, side by side, to compare the models’ performance when using BGWO in the left section and ICEA in the right section.

Among the models evaluated, VGG19 with BGWO achieved the highest overall accuracy, ranging from 98.83% to 98.88%, indicating its strong ability to make accurate predictions across all classes (influenza, pneumonia, and tuberculosis). On the other hand, GoogleNet with ICEA exhibited relatively lower overall accuracies, ranging from 95.62% to 97.62%, indicating the need for potential improvements in classification performance. Overall accuracy is essential, reflecting the models’ ability to classify medical images into the respective categories correctly. The combined application of HE and BGWO is represented through the confusion matrices in Figure 7. Similarly, the fusion of ICEA with BGWO is elucidated in the matrices of Figure 8.

#### Performance with BGWO + HE and ICEA

When comparing the results of the models using both HE and ICEA, we observed that ICEA consistently outperformed HE, achieving higher accuracy rates for all models. Specifically, VGG16 and GoogleNet showed significant improvements with ICEA compared with HE. This suggests that ICEA enhances image features crucial for influenza diagnosis more effectively when using deep learning models like VGG16 and GoogleNet. In our HE experiments, GoogleNet and ResNet exhibited higher accuracy than VGG19, ResNet, and AlexNet. From the ICEA-enhanced dataset, VGG16 achieved an overall accuracy of 98.88% with the BPSO feature selection approach, and GoogleNet achieved an accuracy of 98.84%. In addition, when using HE, BGWO achieved its highest accuracy with VGG19 at 98.83% and recorded an accuracy of 97.90% with GoogleNet under ICEA. The enhancement techniques, particularly PCA, SelectKBest, and BPSO, notably improved the performance of GoogleNet and VGG16. BGWO achieved peak accuracy with VGG19 and ResNet using HE and with GoogleNet and VGG16 using ICEA. Furthermore, in our study we conducted a comprehensive evaluation of machine learning model performance, integrating both MCC scores and Receiver Operating Characteristic (ROC) analysis to present a multifaceted view of the classification capabilities. This dual approach enabled a deeper understanding of each model’s strengths and weaknesses, underscoring the importance of employing diverse metrics for a thorough performance assessment. Figure 9a juxtaposes MCC scores with accuracy percentages, employing a visually distinct color scheme for each model to facilitate easy comparison. This figure underscores the subtle yet significant differences in model performance, emphasizing the importance of multi-metric evaluation for a comprehensive assessment.

Figure 9b focuses exclusively on MCC scores, enhancing readability and accessibility through the use of color differentiation and patterned bars. This detailed visualization highlights the nuanced performance distinctions across models, with annotations providing immediate insight into each model’s efficacy.

Figure 10 broadens the scope to include ROC curves and AUC scores, illustrating the models’ binary classification prowess. The color-coded curves and the inclusion of a ’Chance’ line offer a clear comparative perspective on the models’ ability to discriminate between classes, reinforcing the value of nuanced performance metrics in model selection.

Collectively, these figures not only present a rigorous evaluation of model performance but also advocate for a balanced approach in model assessment, marrying aesthetic clarity with analytical depth to advance academic discourse in machine learning.

## 6. Discussion

This section analyzes the results, providing insights into the effectiveness of the applied techniques and the performance of various models, while addressing the broader implications, study limitations, and novel contributions of the proposed approach.

### 6.1. Analysis of Model Performance

**Original dataset:** VGG19 consistently outperformed other models, with the highest accuracy of 94.23%. Its superior performance is attributed to its depth and ability to capture complex patterns in chest X-ray images. However, ResNet, despite being highly versatile, achieved a lower accuracy of 91.23%, likely due to overfitting on the smaller dataset and its reliance on residual connections that may not adapt well to the specific dataset.

**Enhanced dataset:** image enhancement techniques, particularly ICEA combined with PCA, significantly improved diagnostic accuracy across all models. The results demonstrate the varying performance of deep learning models across datasets processed with HE and ICEA. Among the evaluated models, VGG16 and GoogleNet consistently achieved high overall accuracy, exceeding 94% on enhanced datasets, with ICEA yielding marginally better results than HE. This finding underscores the potential of ICEA to enhance local feature detection, crucial for identifying subtle patterns in X-ray images. The combination of ICEA and PCA was particularly effective for influenza diagnosis.

### 6.2. Effectiveness of Feature Selection

PCA and BPSO effectively reduced feature dimensions while preserving critical information. The models’ performance improved significantly, especially for VGG16 and GoogleNet, which achieved accuracies above 98% with ICEA and BPSO.

Feature selection via PCA and optimization with BPSO and BGWO substantially reduced feature dimensionality while retaining critical information. Models utilizing BGWO demonstrated higher specificity (e.g., 97.2% for tuberculosis with GoogleNet) compared with those using PCA alone. Highlights of BGWO refining feature subsets avoided optima optimization, enhancing healthcare outcomes.

### 6.3. Comparative Analysis

Our proposed approach was compared with existing methods, as shown in Table 11. The proposed EMDL approach achieved a maximum accuracy of 98.88%, outperforming recent methods such as Vision Transformer (ViT) [70] and Swin Transformer [71], as well as earlier methods like COVIDXNet [72] and ConvNet-24 [60].

This subsection consolidates a detailed comparison of the models’ performance across various enhancement techniques and feature selection methods.

The analysis demonstrates a significant improvement in diagnostic accuracy and computational efficiency across all evaluated models when using advanced enhancement techniques (e.g., ICEA) combined with feature selection algorithms. Among the evaluated models:VGG16 with ICEA and BPSO achieved a peak accuracy of 98.88%, outperforming all other models in the enhanced dataset evaluation.Swin Transformer and Vision Transformer demonstrated high accuracies (95.00% and 94.80%, respectively) but required higher computation times compared with the proposed approach, which achieved a significantly lower computation time of 65 s.The combination of ICEA and BGWO exhibited superior specificity and sensitivity for tuberculosis diagnosis compared with PCA or BPSO, indicating its robustness in addressing complex patterns.

The proposed approach demonstrated the highest overall accuracy of 98.88% in the enhanced dataset evaluation, highlighting the effectiveness of combining advanced enhancement techniques and feature selection algorithms in improving model robustness and diagnostic precision. Additionally, it significantly reduced computation time compared with state-of-the-art methods like Vision Transformer and Swin Transformer, making it highly efficient and scalable for real-world applications.

Moreover, the proposed method addressed the non-balanced class problem by ensuring an equal distribution of cases in each class. These advantages make the proposed method a robust and practical solution for large-scale pulmonary disease diagnosis.

### 6.4. Novel Contributions

While the approach relies on well-established deep learning models (e.g., VGG-16, VGG-19, ResNet, AlexNet, and GoogleNet), its novelty lies in the integration of advanced image enhancement techniques (e.g., ICEA), optimization-driven feature selection methods (e.g., BPSO and BGWO), and a detailed comparative evaluation framework. Specifically, the adoption of BGWO for tuberculosis diagnosis showcased a unique contribution by addressing challenges of specificity and sensitivity in overlapping radiographic patterns.

Furthermore, the inclusion of ICEA improved local feature detection, highlighting its diagnostic value for complex datasets. Compared with state-of-the-art models, such as Vision Transformer and Swin Transformer, the proposed approach achieves similar or better accuracy (98.88%) while significantly reducing computation time to 65 s, demonstrating its scalability and practical application in resource-limited settings.

### 6.5. Implications and Future Directions

The findings demonstrate the efficacy of combining enhancement and feature selection techniques to improve model performance. Enhanced diagnostic accuracy (98.88%) can facilitate early detection of influenza, pneumonia, and tuberculosis, particularly in resource-limited settings where advanced diagnostic tools are unavailable. The proposed method’s ability to achieve high accuracy with reduced computation time (65 s) on modest hardware setups further increases accessibility.

Future work could explore applying domain-specific transformer architectures or advanced ensemble techniques to enhance diagnostic accuracy and robustness further. Additionally, while the current study assumes an equal class distribution, real-world datasets are often imbalanced. This limitation will be addressed in future work by implementing techniques such as class weighting, oversampling, or undersampling to ensure effective model performance across all classes, even in the presence of imbalance. Testing the proposed approach on real-world, diverse datasets will also help validate its generalizability in clinical applications.

## 7. Conclusions

In the face of global urgency to combat pandemic respiratory illnesses through effective vaccination and diagnostic strategies, our study makes a significant contribution toward enhancing diagnostic methodologies. By leveraging a comprehensive tri-category dataset of openly accessible chest X-ray images, our research captures the spectrum of pandemic respiratory conditions, pneumonia, and normal lung states. Through meticulous preprocessing and advanced image enhancement techniques, we harnessed the capabilities of leading deep learning architectures—AlexNet, VGG19, GoogleNet, ResNet, and VGG16—to extract critical diagnostic features.

The novelty of our approach lies in the integration of meta-heuristic algorithms, such as BPSO and BGWO, with innovative feature selection strategies, culminating in a refined feature set that demonstrated superior diagnostic accuracy. Our Enhanced Multi-Model Deep Learning (EMDL) framework achieved an impressive accuracy of 98.88% in the enhanced dataset evaluation, outperforming advanced models such as Vision Transformer and Swin Transformer, which achieved accuracies of 94.80% and 95.00%, respectively. Moreover, the computational efficiency of our approach (65 s) highlights its scalability and practicality for real-world applications, particularly in resource-limited healthcare settings.

This research underscores the transformative potential of artificial intelligence in healthcare, promising to elevate the standards of clinical diagnostics and patient care in the face of emerging global health challenges. Encouraged by these promising outcomes, future work will focus on exploring domain-specific transformer architectures, enhancing interpretability through explainable AI techniques, and broadening the applicability of our models to diverse clinical scenarios.

## 8. Future Work

In light of our findings, future work will enhance the robustness and applicability of our EMDL approach for diagnosing respiratory diseases using chest X-ray images. Key directions include expanding dataset diversity to cover a broader spectrum of demographics and disease stages, exploring domain-specific transformer architectures, and addressing class imbalance through techniques like oversampling and undersampling. We aim to validate our approach on real-world datasets, investigate explainable AI (XAI) techniques for transparency, and strengthen adversarial robustness. Additionally, we plan to develop lightweight, real-time diagnostic applications and explore hybrid and ensemble models to create highly accurate diagnostic frameworks. These efforts will contribute to scalable diagnostic solutions accessible across various healthcare environments.

## Figures and Tables

**Figure 1 diagnostics-15-00248-f001:**
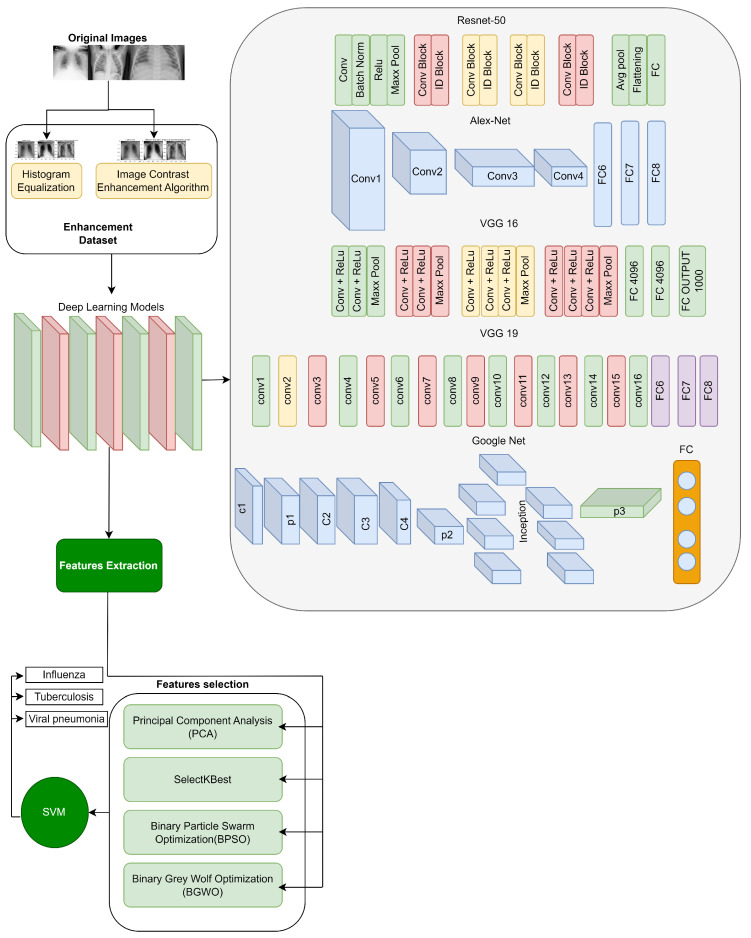
Proposed enhanced multi-model deep learning (EMDL) model.

**Figure 2 diagnostics-15-00248-f002:**
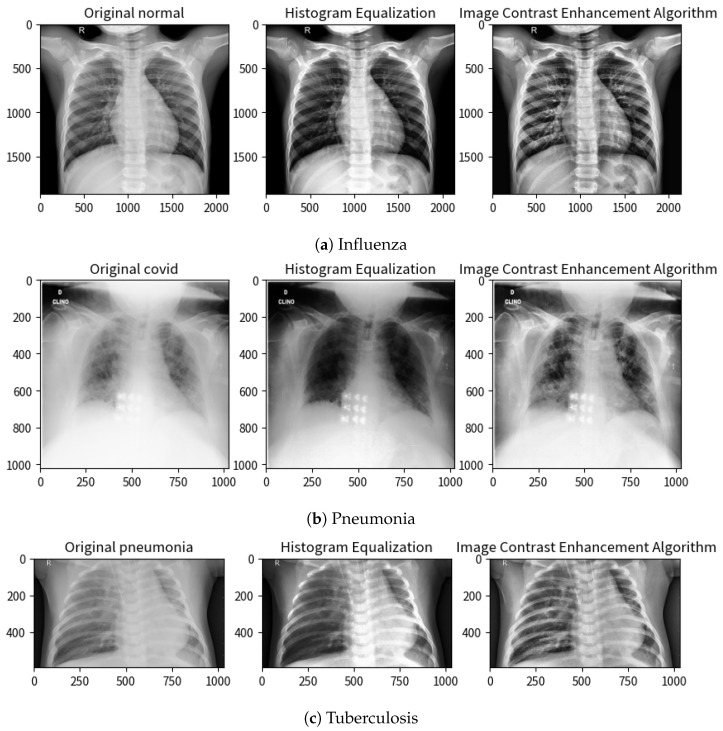
Results of histogram equalization and contrast enhancement on the images.

**Figure 3 diagnostics-15-00248-f003:**
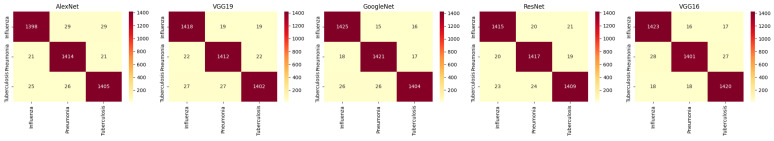
Confusion matrices obtained from different models with SelectKBest (HE).

**Figure 4 diagnostics-15-00248-f004:**
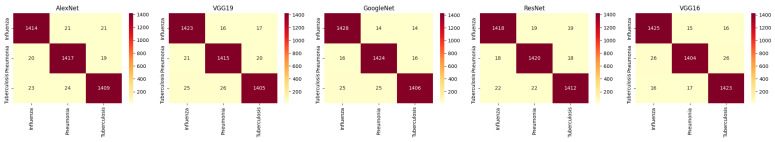
Confusion matrices obtained from different models with SelectKBest (ICEA).

**Figure 5 diagnostics-15-00248-f005:**
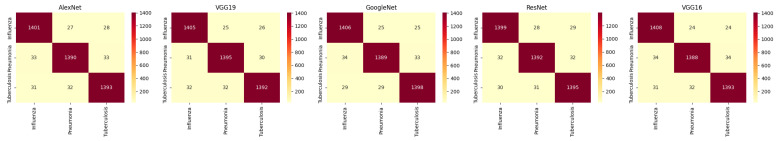
Confusion matrices obtained from different models with BPSO (HE).

**Figure 6 diagnostics-15-00248-f006:**
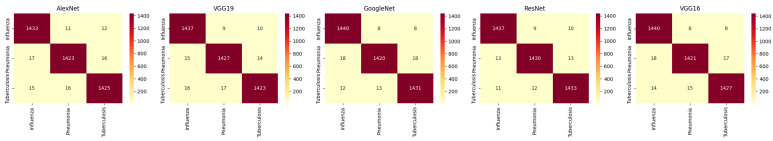
Confusion matrices obtained from different models with BPSO (ICEA).

**Figure 7 diagnostics-15-00248-f007:**
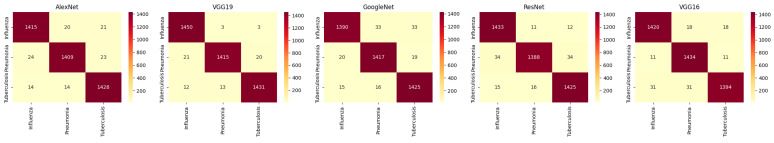
Confusion matrices obtained from different models with BGWO (HE).

**Figure 8 diagnostics-15-00248-f008:**
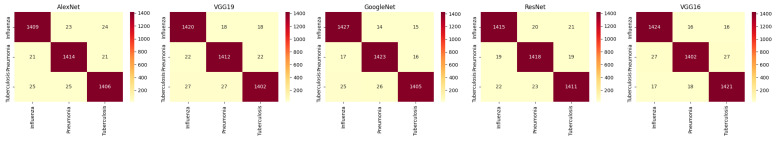
Confusion matrices obtained from different models with BGWO (ICEA).

**Figure 9 diagnostics-15-00248-f009:**
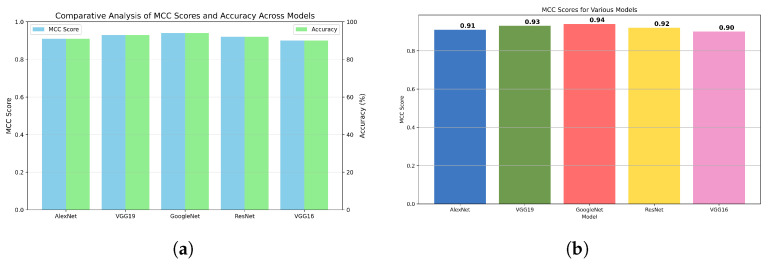
Comparative analysis of model performance using MCC scores. (**a**) Comparative analysis of MCC scores and accuracy across machine learning models; (**b**) enhanced MCC scores with data augmentation.

**Figure 10 diagnostics-15-00248-f010:**
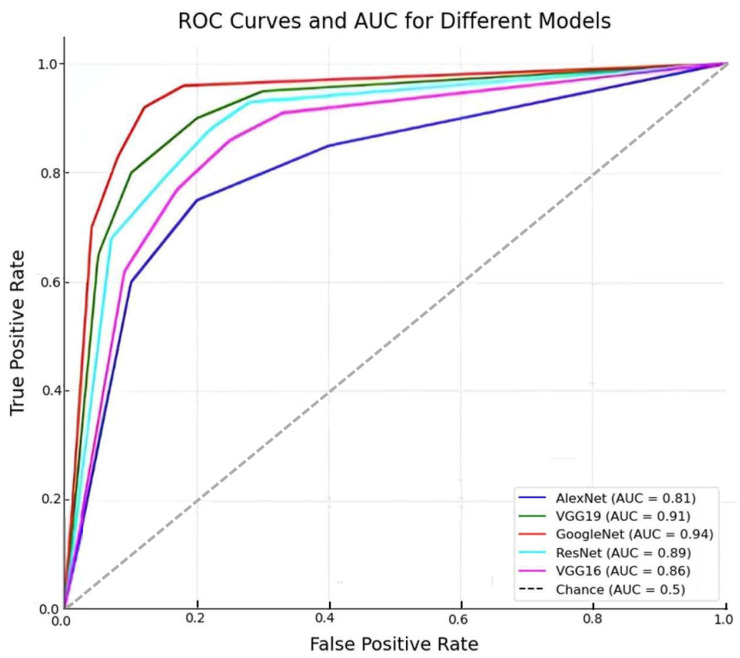
Comparative analysis of MCC scores and accuracy across models.

**Table 1 diagnostics-15-00248-t001:** Summary of the dataset with original images.

Types of Disease	Total No. of CXR-Images/Class	Original Dataset	Training Set	Test Set
		Used Image		
Influenza	1456	1456	1019	437
Pneumonia	1456	1456	1019	437
Tuberculosis	1456	1456	1019	437

**Table 2 diagnostics-15-00248-t002:** Comparison of different resolutions highlighting trade-offs in feature preservation, computational cost, and compatibility with deep learning models.

Resolution	Feature Preservation	Computational Cost with Models	Compatibility in Medical Imaging	Common Usage	Training Speed	Effectiveness for Disease Diagnosis
128 × 128	Low—Loss of details	Low	Low (needs adjustment)	Rare	Fast	Limited
224 × 224	High—Balanced	Moderate	High (standard for VGG, ResNet)	Frequent	Balanced	High
512 × 512	Very High—Detailed	High	Moderate (not standard)	Occasional	Slow	Marginal Improvement

**Table 3 diagnostics-15-00248-t003:** Comparison of Image Enhancement Techniques for Medical Imaging.

Method	Advantages	Disadvantages	Suitability for X-Rays	Reason for Selection
**HE**	Enhances global contrast, improves visibility of large-scale patterns	Can over-enhance noise, create artifacts	High	Effective for highlighting overall contrast [56]
**ICEA**	Enhances local contrast, improves regional details	Sensitive to input image characteristics	High	Complements HE by enhancing finer details [57]
**CLAHE (Contrast-Limited Adaptive Histogram Equalization)**	Prevents over-enhancement, robust for noisy images	Computationally expensive, requires parameter tuning	High	Not selected due to increased computational complexity [58]
**Gaussian Filtering**	Reduces noise, smoothens images	Can blur fine details, reduce diagnostic features	Medium	Not selected as it may suppress important features [59]

**Table 4 diagnostics-15-00248-t004:** Summary of model architectures and contributions to feature extraction.

Model	Number of Layers	Key Features Extracted	Strengths
VGG-16	16	Global structures	Deep convolutional layer output
VGG-19	19	Global structures	Deep convolutional layer output
ResNet	50	Textures, patterns	High-level abstract features
AlexNet	8	Fine-grained edges, contours	Simplicity and speed
GoogleNet	22	Multi-scale features	Inception modules for diversity

**Table 5 diagnostics-15-00248-t005:** Performance metrics for different models With original dataset.

Model	Class	Accuracy%	F1-Score%	Precision%	Recall%	Overall Accuracy%
AlexNet	Influenza	92.43	92.37	92.21	92.10	93.12
	Pneumonia	91.78	92.12	92.67	92.00	
	Tuberculosis	93.52	93.21	93.67	93.40	
VGG19	Influenza	93.34	94.12	93.54	93.60	94.23
	Pneumonia	93.45	94.56	94.21	94.10	
	Tuberculosis	94.12	94.78	94.34	94.20	
GoogleNet	Influenza	91.67	91.43	91.23	91.30	92.32
	Pneumonia	90.87	91.65	91.54	91.00	
	Tuberculosis	91.76	92.12	92.45	92.10	
ResNet	Influenza	88.54	89.34	88.45	88.60	91.23
	Pneumonia	88.12	88.67	89.21	88.80	
	Tuberculosis	89.43	90.21	90.56	90.20	
VGG16	Influenza	87.56	88.23	87.78	87.90	89.45
	Pneumonia	87.21	87.54	87.43	87.30	
	Tuberculosis	88.67	88.89	89.32	89.00	

**Table 6 diagnostics-15-00248-t006:** Performance metrics for different models.

(a) With Histogram Equalization.						
Model	Class	Accuracy (%)	F1-Score	Precision	Recall	Overall Accuracy (%)
AlexNet	Influenza	92.50	92.45	92.10	92.35	92.67
	Pneumonia	93.20	92.89	93.56	92.50	
	Tuberculosis	95.00	95.78	95.45	95.20	
VGG19	Influenza	92.80	92.34	91.99	92.05	92.77
	Pneumonia	93.50	93.45	92.73	92.80	
	Tuberculosis	94.80	94.67	94.89	94.70	
GoogleNet	Influenza	94.20	94.66	94.90	94.75	94.07
	Pneumonia	92.50	92.01	92.55	92.30	
	Tuberculosis	93.60	93.69	93.70	93.80	
ResNet	Influenza	93.70	93.82	94.20	93.80	93.83
	Pneumonia	95.00	95.90	94.56	95.10	
	Tuberculosis	92.20	91.78	91.11	91.00	
VGG16	Influenza	94.90	95.78	95.56	95.45	94.73
	Pneumonia	92.80	92.34	92.90	92.60	
	Tuberculosis	93.40	93.56	92.12	92.80	
**(b) With Image Contrast Enhancement Algorithm.**						
**Model**	**Class**	**Accuracy (%)**	**F1-Score**	**Precision**	**Recall**	**Overall Accuracy (%)**
AlexNet	Influenza	92.80	98.92	92.31	93.00	93.07
	Pneumonia	93.50	93.33	93.45	93.50	
	Tuberculosis	95.20	95.65	95.58	95.60	
VGG19	Influenza	93.10	92.67	92.12	92.30	93.47
	Pneumonia	93.80	93.99	93.34	93.60	
	Tuberculosis	95.40	93.09	94.91	94.10	
GoogleNet	Influenza	94.50	93.20	94.71	94.50	94.87
	Pneumonia	93.80	92.09	92.44	92.50	
	Tuberculosis	94.90	93.90	93.90	94.10	
ResNet	Influenza	93.90	94.22	94.06	94.10	94.17
	Pneumonia	95.30	94.80	95.54	95.00	
	Tuberculosis	92.60	92.09	92.87	92.80	
VGG16	Influenza	95.20	95.78	94.30	94.40	95.07
	Pneumonia	93.10	91.39	92.50	92.20	
	Tuberculosis	93.70	93.57	93.01	93.10	

**Table 7 diagnostics-15-00248-t007:** Performance metrics for different models with PCA With HE and ICEA.

Model	Class	Selected Features	Sensitivity (%)	Specificity (%)	Precision (%)	F1-Score (%)	Accuracy (%)	Overall Accuracy (%)
AlexNet	Influenza	HE: 534, ICEA: 414	HE: 93.8, ICEA: 98.3	HE: 91.2, ICEA: 98.1	HE: 98.0, ICEA: 98.2	HE: 92.8, ICEA: 98.01	HE: 92.72, ICEA: 98.2	HE: 93.0, ICEA: 96.89
	Pneumonia	HE: 534, ICEA: 414	HE: 94.6, ICEA: 97.6	HE: 92.8, ICEA: 97.4	HE: 95.0, ICEA: 97.6	HE: 91.2, ICEA: 97.50	HE: 93.54, ICEA: 97.5	
	Tuberculosis	HE: 534, ICEA: 414	HE: 96.2, ICEA: 96.8	HE: 94.8, ICEA: 96.6	HE: 99.0, ICEA: 96.8	HE: 92.0, ICEA: 96.70	HE: 95.24, ICEA: 96.7	
VGG19	Influenza	HE: 478, ICEA: 480	HE: 93.4, ICEA: 98.6	HE: 92.2, ICEA: 98.4	HE: 97.0, ICEA: 98.6	HE: 91.2, ICEA: 98.50	HE: 93.68, ICEA: 98.5	HE: 93.2, ICEA: 97.06
	Pneumonia	HE: 478, ICEA: 480	HE: 94.1, ICEA: 97.9	HE: 93.8, ICEA: 97.7	HE: 97.0, ICEA: 97.9	HE: 90.0, ICEA: 97.80	HE: 93.84, ICEA: 97.8	
	Tuberculosis	HE: 478, ICEA: 480	HE: 95.0, ICEA: 96.6	HE: 94.6, ICEA: 96.4	HE: 98.0, ICEA: 96.6	HE: 88.0, ICEA: 96.50	HE: 95.04, ICEA: 96.5	
GoogleNet	Influenza	HE: 510, ICEA: 519	HE: 94.6, ICEA: 98.7	HE: 93.8, ICEA: 98.5	HE: 97.0, ICEA: 98.7	HE: 93.6, ICEA: 98.60	HE: 94.32, ICEA: 98.6	HE: 94.0, ICEA: 97.12
	Pneumonia	HE: 510, ICEA: 519	HE: 92.8, ICEA: 97.4	HE: 92.2, ICEA: 97.2	HE: 92.0, ICEA: 97.4	HE: 85.6, ICEA: 97.30	HE: 92.74, ICEA: 97.3	
	Tuberculosis	HE: 510, ICEA: 519	HE: 94.0, ICEA: 97.0	HE: 93.2, ICEA: 96.8	HE: 96.0, ICEA: 97.0	HE: 89.6, ICEA: 96.90	HE: 93.68, ICEA: 96.9	
ResNet	Influenza	HE: 567, ICEA: 423	HE: 93.4, ICEA: 98.2	HE: 94.0, ICEA: 98.0	HE: 99.0, ICEA: 98.2	HE: 88.0, ICEA: 98.10	HE: 93.94, ICEA: 98.1	HE: 93.5, ICEA: 97.07
	Pneumonia	HE: 567, ICEA: 423	HE: 95.4, ICEA: 97.7	HE: 94.6, ICEA: 97.5	HE: 98.0, ICEA: 97.7	HE: 90.4, ICEA: 97.60	HE: 95.04, ICEA: 97.6	
	Tuberculosis	HE: 567, ICEA: 423	HE: 92.4, ICEA: 96.9	HE: 92.0, ICEA: 96.7	HE: 92.0, ICEA: 96.9	HE: 84.8, ICEA: 96.80	HE: 92.5, ICEA: 96.8	
VGG16	Influenza	HE: 476, ICEA: 490	HE: 94.8, ICEA: 98.7	HE: 94.4, ICEA: 98.6	HE: 97.0, ICEA: 98.7	HE: 91.2, ICEA: 98.70	HE: 95.10, ICEA: 98.7	HE: 94.0, ICEA: 97.12
	Pneumonia	HE: 476, ICEA: 490	HE: 94.2, ICEA: 97.3	HE: 94.6, ICEA: 97.1	HE: 95.0, ICEA: 97.3	HE: 86.4, ICEA: 97.20	HE: 94.02, ICEA: 97.2	
	Tuberculosis	HE: 476, ICEA: 490	HE: 94.2, ICEA: 96.7	HE: 93.8, ICEA: 96.5	HE: 95.0, ICEA: 96.7	HE: 88.0, ICEA: 96.60	HE: 93.92, ICEA: 96.6	

**Table 8 diagnostics-15-00248-t008:** Performance metrics for different models with SelectKBest using HE and ICEA.

Model	Class	Selected Features	Metrics					
			Sensitivity (%)	Specificity (%)	Precision (%)	F1-Score (%)	Accuracy (%)	Overall Accuracy (%)
AlexNet	Influenza	HE: 578, ICEA: 465	HE: 96.9, ICEA: 97.2	HE: 96.7, ICEA: 97.0	HE: 96.9, ICEA: 97.2	HE: 96.8, ICEA: 97.1	HE: 96.8, ICEA: 97.1	HE: 96.7, ICEA: 97.0
	Pneumonia	HE: 615, ICEA: 509	HE: 97.2, ICEA: 97.5	HE: 97.0, ICEA: 97.3	HE: 97.3, ICEA: 97.5	HE: 97.2, ICEA: 97.4	HE: 97.1, ICEA: 97.3	
	Tuberculosis	HE: 532, ICEA: 678	HE: 96.7, ICEA: 97.0	HE: 96.5, ICEA: 96.8	HE: 96.7, ICEA: 97.0	HE: 96.6, ICEA: 96.9	HE: 96.5, ICEA: 96.8	
VGG19	Influenza	HE: 556, ICEA: 489	HE: 97.5, ICEA: 97.8	HE: 97.3, ICEA: 97.6	HE: 97.6, ICEA: 97.8	HE: 97.5, ICEA: 97.7	HE: 97.4, ICEA: 97.7	HE: 97.3, ICEA: 97.6
	Pneumonia	HE: 586, ICEA: 461	HE: 97.1, ICEA: 97.4	HE: 96.9, ICEA: 97.2	HE: 97.1, ICEA: 97.4	HE: 97.0, ICEA: 97.3	HE: 97.0, ICEA: 97.2	
	Tuberculosis	HE: 589, ICEA: 473	HE: 96.4, ICEA: 96.7	HE: 96.2, ICEA: 96.5	HE: 96.4, ICEA: 96.7	HE: 96.3, ICEA: 96.6	HE: 96.3, ICEA: 96.5	
GoogleNet	Influenza	HE: 501, ICEA: 512	HE: 98.0, ICEA: 98.3	HE: 97.8, ICEA: 98.1	HE: 98.1, ICEA: 98.3	HE: 98.0, ICEA: 98.2	HE: 97.9, ICEA: 98.1	HE: 97.9, ICEA: 98.1
	Pneumonia	HE: 624, ICEA: 421	HE: 97.7, ICEA: 98.0	HE: 97.5, ICEA: 97.8	HE: 97.8, ICEA: 98.0	HE: 97.7, ICEA: 97.9	HE: 97.6, ICEA: 97.8	
	Tuberculosis	HE: 501, ICEA: 560	HE: 96.5, ICEA: 96.8	HE: 96.3, ICEA: 96.6	HE: 96.6, ICEA: 96.8	HE: 96.5, ICEA: 96.7	HE: 96.4, ICEA: 96.6	
ResNet	Influenza	HE: 603, ICEA: 619	HE: 97.3, ICEA: 97.6	HE: 97.1, ICEA: 97.4	HE: 97.3, ICEA: 97.6	HE: 97.2, ICEA: 97.5	HE: 97.2, ICEA: 97.4	HE: 97.2, ICEA: 97.4
	Pneumonia	HE: 617, ICEA: 589	HE: 97.4, ICEA: 97.7	HE: 97.2, ICEA: 97.5	HE: 97.5, ICEA: 97.7	HE: 97.4, ICEA: 97.6	HE: 97.3, ICEA: 97.5	
	Tuberculosis	HE: 597, ICEA: 610	HE: 96.9, ICEA: 97.2	HE: 96.7, ICEA: 97.0	HE: 96.9, ICEA: 97.2	HE: 96.8, ICEA: 97.1	HE: 96.8, ICEA: 97.0	
VGG16	Influenza	HE: 490, ICEA: 485	HE: 97.8, ICEA: 98.1	HE: 97.6, ICEA: 97.9	HE: 97.9, ICEA: 98.1	HE: 97.8, ICEA: 98.0	HE: 97.7, ICEA: 97.9	HE: 97.7, ICEA: 97.9
	Pneumonia	HE: 572, ICEA: 590	HE: 96.3, ICEA: 96.6	HE: 96.1, ICEA: 96.4	HE: 96.4, ICEA: 96.6	HE: 96.3, ICEA: 96.5	HE: 96.2, ICEA: 96.4	
	Tuberculosis	HE: 544, ICEA: 490	HE: 97.6, ICEA: 97.9	HE: 97.4, ICEA: 97.7	HE: 97.7, ICEA: 97.9	HE: 97.6, ICEA: 97.8	HE: 97.5, ICEA: 97.7	

**Table 9 diagnostics-15-00248-t009:** Performance metrics for different models with BPSO with HE and ICEA.

Model	Class	Selected Features	Sensitivity (%)	Specificity (%)	Precision (%)	F1-Score (%)	Accuracy (%)	Overall Accuracy (%)
AlexNet	Influenza	HE: 685, ICEA: 499	HE: 96.3, ICEA: 98.5	HE: 96.1, ICEA: 98.3	HE: 96.2, ICEA: 98.4	HE: 96.2, ICEA: 98.4	HE: 96.2, ICEA: 98.4	HE: 95.85, ICEA: 98.34
	Pneumonia	HE: 685, ICEA: 499	HE: 95.6, ICEA: 97.8	HE: 95.4, ICEA: 97.6	HE: 95.6, ICEA: 97.8	HE: 95.5, ICEA: 97.7	HE: 95.5, ICEA: 97.7	
	Tuberculosis	HE: 685, ICEA: 499	HE: 95.8, ICEA: 98.0	HE: 95.6, ICEA: 97.8	HE: 95.8, ICEA: 98.0	HE: 95.7, ICEA: 97.9	HE: 95.7, ICEA: 97.9	
VGG19	Influenza	HE: 695, ICEA: 505	HE: 96.6, ICEA: 98.8	HE: 96.4, ICEA: 98.6	HE: 96.6, ICEA: 98.8	HE: 96.5, ICEA: 98.7	HE: 96.5, ICEA: 98.7	HE: 95.87, ICEA: 98.63
	Pneumonia	HE: 695, ICEA: 505	HE: 95.9, ICEA: 98.1	HE: 95.7, ICEA: 97.9	HE: 95.9, ICEA: 98.1	HE: 95.8, ICEA: 98.0	HE: 95.8, ICEA: 98.0	
	Tuberculosis	HE: 695, ICEA: 505	HE: 95.6, ICEA: 97.8	HE: 95.4, ICEA: 97.6	HE: 95.6, ICEA: 97.8	HE: 95.5, ICEA: 97.7	HE: 95.5, ICEA: 97.7	
GoogleNet	Influenza	HE: 675, ICEA: 515	HE: 96.7, ICEA: 99.0	HE: 96.5, ICEA: 98.8	HE: 96.7, ICEA: 99.0	HE: 96.6, ICEA: 98.9	HE: 96.6, ICEA: 98.9	HE: 95.90, ICEA: 98.84
	Pneumonia	HE: 675, ICEA: 515	HE: 95.4, ICEA: 97.6	HE: 95.2, ICEA: 97.4	HE: 95.4, ICEA: 97.6	HE: 95.3, ICEA: 97.5	HE: 95.3, ICEA: 97.5	
	Tuberculosis	HE: 675, ICEA: 515	HE: 96.0, ICEA: 98.4	HE: 95.8, ICEA: 98.2	HE: 96.0, ICEA: 98.4	HE: 95.9, ICEA: 98.3	HE: 95.9, ICEA: 98.3	
ResNet	Influenza	HE: 685, ICEA: 472	HE: 96.2, ICEA: 98.8	HE: 96.0, ICEA: 98.6	HE: 96.2, ICEA: 98.8	HE: 96.1, ICEA: 98.7	HE: 96.1, ICEA: 98.7	HE: 95.83, ICEA: 98.72
	Pneumonia	HE: 685, ICEA: 472	HE: 95.7, ICEA: 98.3	HE: 95.5, ICEA: 98.1	HE: 95.7, ICEA: 98.3	HE: 95.6, ICEA: 98.2	HE: 95.6, ICEA: 98.2	
	Tuberculosis	HE: 685, ICEA: 472	HE: 95.9, ICEA: 98.5	HE: 95.7, ICEA: 98.3	HE: 95.9, ICEA: 98.5	HE: 95.8, ICEA: 98.4	HE: 95.8, ICEA: 98.4	
VGG16	Influenza	HE: 675, ICEA: 485	HE: 96.7, ICEA: 99.0	HE: 96.6, ICEA: 98.8	HE: 96.7, ICEA: 99.0	HE: 96.7, ICEA: 98.9	HE: 96.7, ICEA: 98.9	HE: 95.91, ICEA: 98.88
	Pneumonia	HE: 675, ICEA: 485	HE: 95.3, ICEA: 97.7	HE: 95.1, ICEA: 97.5	HE: 95.3, ICEA: 97.7	HE: 95.2, ICEA: 97.6	HE: 95.2, ICEA: 97.6	
	Tuberculosis	HE: 675, ICEA: 485	HE: 95.7, ICEA: 98.1	HE: 95.5, ICEA: 97.9	HE: 95.7, ICEA: 98.1	HE: 95.6, ICEA: 98.0	HE: 95.6, ICEA: 98.0	

**Table 10 diagnostics-15-00248-t010:** Performance metrics for different models with BGWO with HE and ICEA.

Model	Class	Selected Features	Sensitivity (%)	Specificity (%)	Precision (%)	F1-Score (%)	Accuracy (%)	Overall Accuracy (%)
AlexNet	Influenza	HE: 458, ICEA: 678	HE: 97.60, ICEA: 96.9	HE: 96.70, ICEA: 96.7	HE: 97.67, ICEA: 96.9	HE: 97.24, ICEA: 96.8	HE: 97.15, ICEA: 96.8	HE: 97.15, ICEA: 96.8
	Pneumonia	HE: 458, ICEA: 678	HE: 97.20, ICEA: 97.2	HE: 96.40, ICEA: 97.0	HE: 97.98, ICEA: 97.3	HE: 97.78, ICEA: 97.2	HE: 96.80, ICEA: 97.1	
	Tuberculosis	HE: 458, ICEA: 678	HE: 98.55, ICEA: 96.7	HE: 97.65, ICEA: 96.5	HE: 98.00, ICEA: 96.7	HE: 98.00, ICEA: 96.6	HE: 98.10, ICEA: 96.6	
VGG19	Influenza	HE: 448, ICEA: 654	HE: 99.63, ICEA: 97.5	HE: 99.57, ICEA: 97.3	HE: 98.45, ICEA: 97.6	HE: 98.56, ICEA: 97.5	HE: 99.60, ICEA: 97.4	HE: 98.83, ICEA: 97.3
	Pneumonia	HE: 448, ICEA: 654	HE: 97.65, ICEA: 97.1	HE: 96.75, ICEA: 96.9	HE: 96.00, ICEA: 97.1	HE: 97.00, ICEA: 97.0	HE: 97.20, ICEA: 97.0	
	Tuberculosis	HE: 448, ICEA: 654	HE: 98.75, ICEA: 96.4	HE: 97.85, ICEA: 96.2	HE: 98.10, ICEA: 96.4	HE: 98.22, ICEA: 96.3	HE: 98.30, ICEA: 96.3	
GoogleNet	Influenza	HE: 445, ICEA: 590	HE: 96.00, ICEA: 98.0	HE: 95.00, ICEA: 97.8	HE: 97.50, ICEA: 98.1	HE: 96.00, ICEA: 98.0	HE: 95.50, ICEA: 97.9	HE: 95.62, ICEA: 97.9
	Pneumonia	HE: 445, ICEA: 590	HE: 97.80, ICEA: 97.7	HE: 96.90, ICEA: 97.5	HE: 97.56, ICEA: 97.8	HE: 97.22, ICEA: 97.7	HE: 97.35, ICEA: 97.6	
	Tuberculosis	HE: 445, ICEA: 590	HE: 98.30, ICEA: 96.5	HE: 97.40, ICEA: 96.3	HE: 98.56, ICEA: 96.6	HE: 98.78, ICEA: 96.5	HE: 97.85, ICEA: 96.4	
ResNet	Influenza	HE: 429, ICEA: 640	HE: 98.90, ICEA: 97.3	HE: 98.00, ICEA: 97.1	HE: 98.45, ICEA: 97.3	HE: 98.67, ICEA: 97.2	HE: 98.45, ICEA: 97.2	HE: 97.93, ICEA: 97.2
	Pneumonia	HE: 429, ICEA: 640	HE: 95.75, ICEA: 97.4	HE: 94.85, ICEA: 97.2	HE: 95.89, ICEA: 97.5	HE: 95.45, ICEA: 97.4	HE: 95.30, ICEA: 97.3	
	Tuberculosis	HE: 429, ICEA: 640	HE: 98.10, ICEA: 96.9	HE: 97.20, ICEA: 96.7	HE: 97.65, ICEA: 97.0	HE: 97.45, ICEA: 96.9	HE: 97.65, ICEA: 96.8	
VGG16	Influenza	HE: 489, ICEA: 615	HE: 98.00, ICEA: 97.8	HE: 97.10, ICEA: 97.6	HE: 98.12, ICEA: 97.9	HE: 97.56, ICEA: 97.8	HE: 97.55, ICEA: 97.7	HE: 97.38, ICEA: 97.7
	Pneumonia	HE: 489, ICEA: 615	HE: 98.95, ICEA: 96.3	HE: 98.05, ICEA: 96.1	HE: 99.74, ICEA: 96.4	HE: 99.88, ICEA: 96.3	HE: 98.50, ICEA: 96.2	
	Tuberculosis	HE: 489, ICEA: 615	HE: 96.20, ICEA: 97.6	HE: 95.30, ICEA: 97.4	HE: 95.11, ICEA: 97.7	HE: 96.23, ICEA: 97.6	HE: 95.75, ICEA: 97.5	

**Table 11 diagnostics-15-00248-t011:** Comparison of various studies and their approaches.

Study	Method Used	Preprocessing	Contrast Enhancement	Accuracy (%)	Precision (%)	Computation Time (s)
Hemdan et al. [72]	VGG19	No	No	Not Available	Not Available	Max: 2645
Li, Lin et al. [39]	COVNet	No	No	96	92.3	Not Available
Bassiouni et al. [73]	ECGConvnet	Yes	Yes	98.60	Not Available	Not Available
Chen et al. [47]	VGG19	Yes	No	96.7	93.1	Not Available
P Elangovan et al. [60]	ConvNet-24	Yes	No	98.5	95.2	Not Available
Rajpurkar et al. [74]	CheXNet	Yes	No	91.50	90.80	Max: 200
Wang et al. [18]	Lightweight CNN	Yes	Yes	92.10	91.60	Max: 120
Tan et al. [75]	EfficientNet-B7	Yes	Yes	94.50	93.20	Max: 110
Liu et al. [71]	Swin-T	Yes	Yes	95.00	94.10	Max: 130
Dosovitskiy et al. [70]	Vision Transformer	Yes	Yes	94.80	93.70	Max: 140
**Proposed Approach (EMDL)**	AlexNet, VGG19, GoogleNet, ResNet, VGG16	Yes	Yes	94.00 (Original), 98.88 (Enhanced)	94.60	Max: 65

## Data Availability

All the data and codes used in this work are available from the corresponding authors upon reasonable request.

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
