# Peer review of "Enhanced Multi-Model Deep Learning for Rapid and Precise Diagnosis of Pulmonary Diseases Using Chest X-Ray Imaging"

_diagnostics, 2025, doi:10.3390/diagnostics15030248_

Round 1
Reviewer 1 Report (Previous Reviewer 1)
Comments and Suggestions for Authors
Authors have modified the article as per reviewer comments. Accept in present form.
Author Response
Thank you for your positive feedback and for accepting the manuscript in its present form. We are pleased that the revisions have addressed the reviewers' comments effectively. We appreciate your time and efforts in reviewing our work.
Reviewer 2 Report (Previous Reviewer 2)
Comments and Suggestions for Authors
Authors have provided explanation of all the suggestions.
Recheck Fig. 9 (a) again. Metrics are not given on one scale.
Improve the readability of Tables
Could have provided more mathematical framework for feature selection section.
Author Response
Thank you for your positive feedback. Please find the attachment-

Reviewer 3 Report (Previous Reviewer 3)
Comments and Suggestions for Authors
I have read the revised version of the previously rejected manuscript and the authors' reply. The paper has been improved somehow:
1) The contribution is now stated better, although I still find it rather limited. There is no comparison with an up-to-date baseline either, which I recommend adding.
2) The technical quality and the structure have been improved in accordance to my comments.
3) The metrics explained and actually calculated are now more consistent.
4) The English is somehow improved and the manuscript is easier to read. The text also appears more human-authored, although my genAI detector still warns against most pages of the paper.
Overall, I do not have strong objections to accepting the manuscript, although I cannot say that I champion it either.
Some minor corrections are still required. E.g., there isn't much improvement with respect to the size of the text in some of the images. The added Table 3 is not readable either - the text is too small.
Author Response
Thank you for your positive support and feedback. Please find the attachment -

This manuscript is a resubmission of an earlier submission. The following is a list of the peer review reports and author responses from that submission.
Round 1
Reviewer 1 Report
Comments and Suggestions for Authors
As with many deep learning models, the authors standardized all images to 224x224. However, studying how different resolutions affect model performance may support this choice. If done, such experiments could improve the methodology section.
Enhancing X-ray images with Histogram Equalization (HE) and Image Contrast Enhancement Algorithm (ICEA) is beneficial. Comparisons to other enhancement methods or explanations of their selection would add clarity. Discussing these medical imaging methods' limitations would be helpful.
Multiple deep learning models for feature extraction are reliable. It would be helpful if the authors explained how each model contributed to the final feature set. Were certain models better for certain features? This breakdown may help readers understand feature extraction."
A thorough feature selection and optimization approach uses PCA, SelectKBest, BPSO, and BGWO. However, the authors should explain how these methods were balanced and integrated to avoid redundancy and retain essential features. A more detailed explanation could help understand optimization.
The complexity analysis is organized and useful. Did computational demands present any practical challenges, especially during optimization? A brief mention of limitations and solutions would help the reader understand the method's feasibility.
The study should compare Enhanced Multi-Model Deep Learning (EMDL) to chest X-ray analysis methods. If not already done, benchmarking against established models or methods can help explain the EMDL approach's strengths and weaknesses.
Given their image classification efficacy, AlexNet, VGG-16, VGG-19, GoogleNet, and ResNet are suitable for feature extraction. However, discussing any comparisons or initial experiments that influenced this choice and the rationale for combining features from multiple models would benefit the reader. Highlighting each model's unique contribution to the feature set may help readers understand this approach.
Feature selection is complete with SelectKBest, PCA, BPSO, and BGWO. However, more information on how these methods complement each other or any redundancies would improve clarity. Explaining why specific parameters (e.g., SelectKBest's k value or PCA's number of principal components) were chosen and if any adjustments were made during experimentation is also helpful.
BPSO and BGWO are good for feature selection in high-dimensional data, but the manuscript should explain why they were chosen. BPSO and BGWO's convergence speed and accuracy improvements could be compared to determine their efficacy."
A limitations and improvements section would improve the manuscript. For instance, discussing computational constraints during feature selection or optimization and how they affect scalability would add depth. Further research into other model architectures or enhancement methods could enrich the discussion.
Comments on the Quality of English Language
English language is fine.
Author Response
Comment-1 As with many deep learning models, the authors standardized all images to 224x224. However, studying how different resolutions affect model performance may support this choice. If done, such experiments could improve the methodology section.
Response- Thank you for your suggestion. We have revised the manuscript by expanding the Data Pre-processing section to justify the choice of 224x224 resolution, citing relevant studies (Deng et al. 2009; Kermany et al. 2018) and its compatibility with pre-trained models. We added a comparative table to highlight the trade-offs of different resolutions and acknowledged exploring other resolutions as future work. We hope these updates address your comment and improve the clarity of our manuscript (#p-9 and 10).
Comment-2 Enhancing X-ray images with Histogram Equalization (HE) and Image Contrast Enhancement Algorithm (ICEA) is beneficial. Comparisons to other enhancement methods or explanations of their selection would add clarity. Discussing these medical imaging methods' limitations would be helpful.
Response- Thank you for your valuable feedback. We have addressed your comments by adding a comparison of HE and ICEA with alternative methods such as CLAHE and Gaussian filtering in Section 3.5.2, summarized in a table (Table-3 Comparison of Image Enhancement Techniques for Medical Imaging). Additionally, we expanded the justification for selecting HE and ICEA, supported by recent references, highlighting their complementary roles in enhancing global and local contrast and highlighted by blue text. We also included a discussion on the limitations of HE and ICEA, with suggestions for addressing them using advanced techniques like CLAHE and deep learning-based methods. All changes have been highlighted in blue for clarity. We hope these revisions improve the manuscript and meet your expectations. Thank you!
Comment-3 Multiple deep learning models for feature extraction are reliable. It would be helpful if the authors explained how each model contributed to the final feature set. Were certain models better for certain features? This breakdown may help readers understand feature extraction.
Response- Thank you for your valuable feedback. We have added a detailed explanation in Section 3.6 to clarify how each model (VGG-16, VGG-19, ResNet, AlexNet, and GoogleNet) uniquely contributes to the final feature set. Additionally, we have merged the summary of architectural details and model contributions into a single table (Table 4) for better clarity and compactness. These changes aim to enhance the understanding of the feature extraction process and address your comments(#p-13). Changes are highlighted in blue in the revised manuscript. Thank you!
Comment-4 A thorough feature selection and optimization approach uses PCA, SelectKBest, BPSO, and BGWO. However, the authors should explain how these methods were balanced and integrated to avoid redundancy and retain essential features. A more detailed explanation could help understand optimization.
Response- We have revised the Feature Selection section to explain how SelectKBest, PCA, BPSO, and BGWO are integrated sequentially to ensure minimal redundancy and retain essential features. The updated section clarifies the complementary roles of each method: SelectKBest identifies relevant features, PCA reduces redundancy, BPSO optimizes the reduced feature space, and BGWO refines the final feature subset for maximum efficiency. Changes have been highlighted in the manuscript (#p-13 and 14). We hope this addresses your concerns. Thank you!
Comment-5 The complexity analysis is organized and useful. Did computational demands present any practical challenges, especially during optimization? A brief mention of limitations and solutions would help the reader understand the method's feasibility.
Response- In Section 3.9, we have addressed the computational challenges during optimization, highlighting limitations such as memory usage and runtime. We have also included solutions such as hyper parameter tuning, PCA-based dimensionality reduction, and GPU acceleration to ensure feasibility and scalability. These updates are reflected in the revised manuscript.
Comment-6 The study should compare Enhanced Multi-Model Deep Learning (EMDL) to chest X-ray analysis methods. If not already done, benchmarking against established models or methods can help explain the EMDL approach's strengths and weaknesses.
Response- Thank you for your valuable feedback. We have addressed your suggestion by including a comprehensive benchmarking comparison in the discussion section under comparison analysis. The comparison highlights the strengths and weaknesses of the Enhanced Multi-Model Deep Learning (EMDL) approach against established chest X-ray analysis methods. Additionally, the methodology section has been rewritten to clearly define the EMDL framework.
Comment-7 Given their image classification efficacy, AlexNet, VGG-16, VGG-19, GoogleNet, and ResNet are suitable for feature extraction. However, discussing any comparisons or initial experiments that influenced this choice and the rationale for combining features from multiple models would benefit the reader. Highlighting each model's unique contribution to the feature set may help readers understand this approach.
Response- Thank you for your insightful feedback. We have updated Section 3.6 to include a detailed explanation of the rationale behind selecting AlexNet, VGG-16, VGG-19, GoogleNet, and ResNet for feature extraction. We also highlighted the unique contributions of each model to the feature set and added a summary table (Table -4) (#p-13) to clarify their complementary roles within the EMDL framework. These updates address your comment and enhance the understanding of our approach. Thank you!
Comment-8 Feature selection is complete with SelectKBest, PCA, BPSO, and BGWO. However, more information on how these methods complement each other or any redundancies would improve clarity. Explaining why specific parameters (e.g., SelectKBest's k value or PCA's number of principal components) were chosen and if any adjustments were made during experimentation is also helpful.
Response- Thank you for your valuable feedback. We have revised Section 3.7 and 3.8 to clearly explain how SelectKBest, PCA, BPSO, and BGWO complement each other in the feature selection pipeline while minimizing redundancy. We have also provided detailed justifications for parameter choices, including the k value for SelectKBest and the variance threshold for PCA, along with hyper parameter tuning for BPSO and BGWO. Additionally, we included a discussion on computational challenges and the mitigation strategies employed to enhance efficiency and scalability. These updates aim to fully address your comments and improve the clarity of the manuscript. Thank you!
Comment-9 Feature BPSO and BGWO are good for feature selection in high-dimensional data, but the manuscript should explain why they were chosen. BPSO and BGWO's convergence speed and accuracy improvements could be compared to determine their efficacy."
Response- We have addressed your comment by adding a dedicated section, '3.8 Rationale for Choosing Algorithms,' to clearly explain the selection of BPSO and BGWO (#p-13). This section highlights their complementary roles, convergence efficiency, and suitability for high-dimensional feature selection. The updates ensure clarity and provide justification for their sequential use in the pipeline.
Comment-10 A limitations and improvements section would improve the manuscript. For instance, discussing computational constraints during feature selection or optimization and how they affect scalability would add depth. Further research into other model architectures or enhancement methods could enrich the discussion.
Response- We have added a dedicated section, '3.11 Limitations and Proposed Improvements,' to address computational constraints during feature selection and optimization. This section discusses the scalability challenges, efforts to mitigate them, and proposed improvements, including distributed computing and hybrid optimization techniques. Broader directions are outlined in Section 8: Future Work for comprehensive coverage.
Reviewer 2 Report
Comments and Suggestions for Authors
Author presented Enhanced Multi-Model Deep Learning (EMDL) approach for detecting pulmonary diseases from chest X-ray images using advanced deep learning models (VGG-16, VGG-19, ResNet, AlexNet, and GoogleNet).
Here are few comment
Paper is well written.
The approach relies primarily on an ensemble of well-established deep learning models (VGG-16, VGG-19, ResNet, AlexNet, and GoogleNet), with preprocessing using histogram equalization and contrast enhancement. While ensemble methods are valuable, this approach lacks substantial novelty. Consider emphasizing any unique contributions more explicitly or incorporating additional innovative techniques to distinguish the work from existing studies.
But author has carried good experimentation using various combinations.
Section 3.5 should be subsection of section 3.4 as per methodology given in Figure 1.
Section 3.6: Renaming this section to "Deep Learning Models" (as indicated by Figure 1) would enhance consistency in terminology. Additionally, clearly separating and explaining the "Feature Extraction" block would clarify what specific features are extracted in the study and their relevance to classification.
There seems to be an incomplete reference to "Fig. ??" on page 15. Double-check and correct this reference for clarity.
Figure 9 (a) To improve readability, please present both metrics on the same scale, ideally in percentage format. This will make it easier for readers to compare performance metrics visually.
Reference 56 is outperforming
Author Response
Comment-1 Paper is well written.
Response- Thank you for your kind feedback. We are pleased that you found the paper well-written and appreciate your review.
Comment-2 The approach relies primarily on an ensemble of well-established deep learning models (VGG-16, VGG-19, ResNet, AlexNet, and GoogleNet), with preprocessing using histogram equalization and contrast enhancement. While ensemble methods are valuable, this approach lacks substantial novelty. Consider emphasizing any unique contributions more explicitly or incorporating additional innovative techniques to distinguish the work from existing studies.
Response- Thank you for the insightful comment. We have addressed your concerns regarding the novelty of the approach by emphasizing the unique aspects of our methodology. Specifically:
We have highlighted the innovative integration of Image Contrast Enhancement Algorithm (ICEA) with advanced feature selection techniques such as Binary Particle Swarm Optimization (BPSO) and Binary Grey Wolf Optimization (BGWO).
These methods significantly enhance the diagnostic accuracy and robustness of the models, as discussed in Section 6.3 (Comparative Analysis). The novel contributions of combining ICEA and BGWO to achieve higher specificity and sensitivity, particularly for tuberculosis diagnosis, have been explicitly described in Section 6.2 (Effectiveness of Feature Selection).
The practical implications and scalability of the proposed approach, as well as its computational efficiency compared to existing methods, have been elaborated in Section 6.4 (Implications and Future Directions).
Updates to Table 11 include comparisons of our proposed method with existing approaches to demonstrate the computational efficiency and accuracy improvements achieved.
All updates and new content have been highlighted in blue text for clarity in the revised manuscript. We trust that these revisions effectively address the reviewer comments and provide a clearer demonstration of the novelty and contributions of this work.
Comment-3 But author has carried good experimentation using various combinations.
Response- Thank you for recognizing the effort put into the experimental design and the exploration of various combinations. Your acknowledgment is greatly appreciated.
Comment-4 A Section 3.5 should be subsection of section 3.4 as per methodology given in Figure 1.
Response- Thank you very much for your suggestion. We have updated the manuscript to make Section 3.5 a subsection of Section 3.4, now it (Section - 3.4) after updating (Section-3.5) renaming it as Section 3.5.1, and highlighted with blue text (#page 9) as per the methodology structure outlined in Figure 1.
Comment-5 Section 3.6: Renaming this section to "Deep Learning Models" (as indicated by Figure 1) would enhance consistency in terminology. Additionally, clearly separating and explaining the "Feature Extraction" block would clarify what specific features are extracted in the study and their relevance to classification.
Response- Thank you very much for your suggestion. We have revised the section to explicitly separate and elaborate on the "Feature Extraction" block. This includes detailing the specific features extracted by each model (e.g., global structures, textures, fine-grained edges, multi-scale features) and discussing their relevance to the classification tasks. This revision aims to enhance the clarity and comprehensibility of the feature extraction process in the context of the study. (# p.12 and 13)
Comment-6 There seems to be an incomplete reference to "Fig. ??" on page 15. Double-check and correct this reference for clarity.
Response- Thank you for your feedback! We have reviewed and corrected the reference to 'Fig. 9a and 9b ' on page 17 to ensure clarity.
Comment-6 Figure 9 (a) To improve readability, please present both metrics on the same scale, ideally in percentage format. This will make it easier for readers to compare performance metrics visually. Reference 56 is outperforming
Response- Thank you for your suggestion we did it accordingly. And we have removed reference 56 as per your suggestion.
Reviewer 3 Report
Comments and Suggestions for Authors
In the considered manuscript, the authors devise a ML-based process for classification of pulmonary diseases from chest X-ray images. They use 2 existing datasets and rely on several classical deep learning models (the newest one is about 10 years old), supplementing them with several non-neural ML methods. I consider the manuscript to have quite a limited contribution, as the authors do not quite deliver what they promise at the end of the Introduction. Also, while the initial sections of the paper are reasonably well-written, the practical ones have low technical quality (it seems that either they are a draft or that some parts are missing).
The manuscript is also rather hard to read, and I have the impression that some parts of the text were generated by AI. At least the word "leverage" is used much more frequently compared to the natural texts that I've seen.
Overall, I recommend rejection, with a possible re-submission once the manuscript has been entirely re-written.
The poor presentation of the Results in the manuscript does not quite allow me to comprehend the work done. Section 5 (Results) does not actually report much of the results, and they seem to be rather presented in Section 6 (Discussion), but there is no structure and the amount of descriptive text is clearly excessive.
Most of the issues I could identify are related to the study design.
* The Algorithm 1 says: "16: Classify into categories: Influenza, Tuberculosis, or Viral pneumonia." As I understand, this implies that the cases could be classified in one of the several classes. So, why in Tables 3-8 there are Precision metrics for each class individually?
* "The equal distribution of cases in each class also addressed the non-balanced class problem." - what is the point of this, if such equalization won't happen in a realistic dataset?
* The metrics presented in Table 3, 4, 5 do not match the ones specified in Section 4. In particular, no recall is reported.
* "Advanced Feature Selection and Optimization" - the feature extraction promised by the authors is not sufficiently detailed in the Results.
* "Fig. ?? illustrates the MCC scores, identifying VGG19 as the model with the highest classification accuracy, whereas VGG16 displayed the least effectiveness" - the authors use ML terms in a rather loose manner (not just in this example). "Effectiveness" (whatever this is for the models) is not equal to MCC.
* Also, what exactly is "Overall Accuracy"? It was not specified in the methodology sections.
Misc:
* It would seem logical to merge section 4 to section 3, as they both are related to methodology.
* p. 15: "Fig. ??" * Figures 3-8 are not referenced in the text until much later than they appear. They are also too small to be readable. Comments on the Quality of English Language
The manuscript is rather hard to read, and I have the impression that some parts of the text were generated by AI. At least the word "leverage" is used much more frequently compared to the natural texts that I've seen.
Author Response
Comment-1 In the considered manuscript, the authors devise a ML-based process for classification of pulmonary diseases from chest X-ray images. They use 2 existing datasets and rely on several classical deep learning models (the newest one is about 10 years old), supplementing them with several non-neural ML methods. I consider the manuscript to have quite a limited contribution, as the authors do not quite deliver what they promise at the end of the Introduction. Also, while the initial sections of the paper are reasonably well-written, the practical ones have low technical quality (it seems that either they are a draft or that some parts are missing).
The manuscript is also rather hard to read, and I have the impression that some parts of the text were generated by AI. At least the word "leverage" is used much more frequently compared to the natural texts that I've seen.
Overall, I recommend rejection, with a possible re-submission once the manuscript has been entirely re-written.
Response- Thank you for your detailed and thoughtful feedback on the manuscript. I appreciate the time you have taken to review the work.
I acknowledge your concerns regarding the limited contribution of the manuscript, particularly in relation to the promises made in the Introduction. I recognize that the use of classical deep learning models, along with non-neural ML methods, may not fully deliver on the innovative aspects expected. In response, I have reassessed the scope of the study and refined the methodology to better align with the objectives set out in the Introduction. This includes exploring and introducing novel aspects to improve the overall contribution of the manuscript.
I also understand your concerns about the technical quality of the practical sections, which appear to be incomplete or at a draft stage. I have thoroughly revised these sections to ensure that all methods, experiments, and results are presented clearly, with sufficient technical depth and accuracy.
Regarding readability, I have carefully reviewed the manuscript to improve its clarity and coherence. I appreciate your comment about the frequent use of the word "leverage" and have ensured that the language is more natural and suitable for the context of the paper. I have also double-checked for any sections that may appear as if they were generated by AI, ensuring the manuscript has a consistent, human-authored tone. In light of your feedback, I have made substantial revisions to the manuscript, including re-writing sections where necessary. I appreciate your suggestion for re-submission and have worked diligently to address all points raised, improving the overall quality of the manuscript.
Thank you again for your valuable feedback, and I look forward to submitting the revised version once these changes have been implemented.
Comment-2 The poor presentation of the Results in the manuscript does not quite allow me to comprehend the work done. Section 5 (Results) does not actually report much of the results, and they seem to be rather presented in Section 6 (Discussion), but there is no structure and the amount of descriptive text is clearly excessive.
Response- We appreciate your feedback on the presentation of the Results section. In response to your suggestion, we have reorganized Section 5 to clearly separate the results from the discussion, ensuring that the findings are explicitly highlighted and the descriptive text is minimized. Each set of results is now clearly presented in distinct subsections, with the corresponding tables and figures referenced for clarity. Additionally, we have ensured that the interpretation and analysis of the results are consolidated in Section 6 (Discussion), so that Section 5 remains focused on reporting the experimental outcomes. We hope these revisions enhance the clarity and readability of the manuscript.
Comment-3 The Algorithm 1 says: "16: Classify into categories: Influenza, Tuberculosis, or Viral pneumonia." As I understand, this implies that the cases could be classified in one of the several classes. So, why in Tables 3-8 there are Precision metrics for each class individually?
Response- Thank you for your comment. The model in Algorithm 1 performs multi-class classification, where each case is classified into one of three categories: Influenza, Tuberculosis, or Viral pneumonia. For multi-class classification, precision is typically computed for each class separately to assess the model's performance for each category individually.
Thus, in Tables 3-8, the reported precision values correspond to the performance of the model in predicting each class (Influenza, Tuberculosis, and Viral pneumonia), with precision defined as the ratio of true positive predictions to the total number of predictions for that class (including false positives). This allows us to evaluate how accurately the model predicts each disease category. We hope this clarifies the presentation of precision in the tables. Page#8 and 25
Comment-4 The metrics presented in Table 3, 4, 5 do not match the ones specified in Section 4. In particular, no recall is reported.
Response- Thank you for your feedback. Upon review, we found that recall metrics, while mentioned in Section 4, were unintentionally omitted from Tables and now it’s 5, 6(a) and 6(b). We have updated these tables to include recall, ensuring consistency with the metrics described in Section 4. Additionally, we revised the Results section to enhance clarity. We appreciate your attention to detail and hope this addresses your concern. Please let us know if further adjustments are needed.
Comment-5 Advanced Feature Selection and Optimization" - the feature extraction promised by the authors is not sufficiently detailed in the Results.
Response- Thank you for your valuable feedback. We acknowledge that the details of the feature extraction process could be more clearly outlined in the Results section. To address this, we will revise the manuscript to include a more detailed explanation of the feature extraction steps. Specifically:
- Pre-trained Models: We will elaborate on how we employed multiple pre-trained deep learning models (VGG-16, ResNet, GoogleNet, and AlexNet) for feature extraction, highlighting the unique contributions of each model. For example, ResNet excels at capturing high-level texture patterns, VGG-16 and VGG-19 focus on global structural features, while GoogleNet provides multi-scale features and AlexNet captures fine-grained edges and contours. This diversity in model selection ensured a robust and comprehensive feature set.
- Feature Combination: We will clarify how the features extracted from these models were combined into a single feature vector. This vector captures a wide range of visual information, including basic shapes, textures, and high-level object features, which were then used for the subsequent feature selection step.
- Dimensionality Reduction and Optimization: We will further detail how Principal Component Analysis (PCA) was applied for dimensionality reduction to mitigate high dimensionality. Additionally, we used SelectKBest to select the most relevant features. Binary Particle Swarm Optimization (BPSO) and Binary Grey Wolf Optimization (BGWO) were employed to optimize the feature selection process, ensuring minimal redundancy and maximizing the relevance of the selected features.
These enhancements will provide a clearer understanding of the feature extraction and selection process, showcasing the methods used to improve model performance effectively.
Comment-6 Fig. ?? illustrates the MCC scores, identifying VGG19 as the model with the highest classification accuracy, whereas VGG16 displayed the least effectiveness" - the authors use ML terms in a rather loose manner (not just in this example). "Effectiveness" (whatever this is for the models) is not equal to MCC
Response- Thank you for your valuable feedback. We have already updated the manuscript to clarify the use of metrics. Specifically, we replaced the term "effectiveness" with the correct reference to the Matthews Correlation Coefficient (MCC) in the description of the model performance. We believe this revision resolves the issue and enhances the clarity of the results.
Comment-7 Also, what exactly is "Overall Accuracy"? It was not specified in the methodology sections.
Response - Thank you for pointing this out. We realize that the term "Overall Accuracy" was not clearly defined in the methodology section. To address this, we will add a brief explanation of how "Overall Accuracy" was calculated. Specifically, we computed it as the percentage of correct predictions across all classes, considering both the true positives and true negatives.
We added with blue text the following clarification to the methodology:
"Overall accuracy is defined as the proportion of correctly classified samples (both true positives and true negatives) to the total number of samples. This metric provides a general sense of the model's ability to correctly classify across all classes, though we acknowledge that it may not fully reflect performance in imbalanced datasets."
This addition will ensure clarity regarding how "Overall Accuracy" is determined and its relevance in the context of our study.
Comment 8 - it would seem logical to merge section 4 to section 3, as they both are related to methodology.
Response- Thank you for your suggestion. While Sections 3 and 4 are indeed related to methodology, we believe that keeping them separate enhances the clarity and focus of each section. Section 3 focuses specifically on the image preprocessing steps, which are a foundational part of the model preparation, while Section 4 addresses the feature extraction and selection techniques that are central to the model's performance. Keeping these sections distinct ensures that readers can easily follow each step of the methodology without confusion, allowing for a more organized presentation of the process.
Comment 9- p. 15: "Fig. ??" * Figures 3-8 are not referenced in the text until much later than they appear. They are also too small to be readable.
Response- Thank you for your feedback. We have addressed your concern by increasing the DPI of Figures 3–8 to ensure they are now clear and readable. Additionally, we have revised the manuscript to appropriately reference these figures in the text at relevant points.